# Discovering Sparsity Allocation for Layer-wise Pruning of Large Language Models

**Lujun Li**[1†]**, Peijie Dong**[2†]**, Zhenheng Tang**[2,3]**, Xiang Liu**[2]**, Qiang Wang**[4]**,**
**Wenhan Luo**[1]**, Wei Xue**[1]**, Qifeng Liu**[1*]**, Xiaowen Chu**[2*]**, Yike Guo**[1*]
[1]Hong Kong University of Science and Technology
[2]Hong Kong University of Science and Technology (Guangzhou)
[3]Hong Kong Baptist University    [4]Harbin Institute of Technology (Shenzhen)
`lilujunai@gmail.com,pdong212@connect.hkust-gz.edu.cn,`
`zhtang@comp.hkbu.edu.hk, xliu886@connect.hkust-gz.edu.cn, qiang.wang@hit.edu.cn,`
`{whluo,weixue,liuqifeng,xwchu,yikeguo}@ust.hk` *

## Abstract

In this paper, we present DSA, the first automated framework for discovering sparsity allocation schemes for layer-wise pruning in Large Language Models (LLMs). LLMs have become increasingly powerful, but their large parameter counts make them computationally expensive. Existing pruning methods for compressing LLMs primarily focus on evaluating redundancies and removing element-wise weights. However, these methods fail to allocate adaptive layer-wise sparsities, leading to performance degradation in challenging tasks. We observe that per-layer importance statistics can serve as allocation indications, but their effectiveness depends on the allocation function between layers. To address this issue, we develop an expression discovery framework to explore potential allocation strategies. Our allocation functions involve two steps: reducing element-wise metrics to per-layer importance scores, and modelling layer importance to sparsity ratios. To search for the most effective allocation function, we construct a search space consisting of pre-process, reduction, transform, and post-process operations. We leverage an evolutionary algorithm to perform crossover and mutation on superior candidates within the population, guided by performance evaluation. Finally, we seamlessly integrate our discovered functions into various uniform methods, resulting in significant performance improvements. We conduct extensive experiments on multiple challenging tasks such as arithmetic, knowledge reasoning, and multimodal benchmarks spanning GSM8K, MMLU, SQA, and VQA, demonstrating that our DSA method achieves significant performance gains on the LLaMA-1|2|3, Mistral, and OPT models. Notably, the LLaMA-1|2|3 model pruned by our DSA reaches 4.73%|6.18%|10.65% gain over the state-of-the-art techniques (*e.g.*, Wanda and SparseGPT).

## 1 Introduction

Large language models (LLMs) [63, 51, 4] have ushered in a new era of natural language processing (NLP) [56], demonstrating remarkable capabilities in understanding and generating human-like text [55]. However, recent LLMs have an incredibly large number of parameters, which contributes to their high computational resource consumption. For example, OpenAI's GPT-3 model has 175 billion parameters and consumed 284,000 kWh of energy during its training [9]. The exponential growth in model size and complexity presents challenges, especially for deployment on resource-constrained

---

**Corresponding authors, † equal contribution. Codes at: https://github.com/lliai/DSA

38th Conference on Neural Information Processing Systems (NeurIPS 2024).

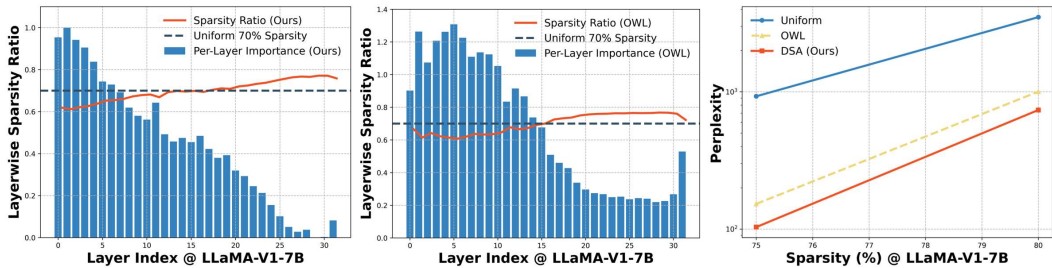

Figure 1: Sparse ratios by our method (***left***) and OWL (***middle***), WikiText-2 perplexity results (***right***).

devices. As a result, there is a pressing need to develop effective compression techniques [3, 67] that can reduce the size of LLMs while preserving their performance. One promising approach is pruning, which involves removing redundant or less important parameters from the model.

Conventional pruning methods [21] propose extensive pruning metrics [61] and sparse training strategies [21]. However, these traditional methods often involve performance drops on small-scale models and require extra fine-tuning, making them difficult to transfer to LLMs due to differences in model structure and the high cost of the extra fine-tuning. To address this, recent approaches like SparseGPT [16] and Wanda [49] have been developed specifically for pruning LLMs. SparseGPT prunes insignificant weights and reconstructs layer-wise outputs based on an importance metric obtained from the Hessian matrix. Wanda proposes a streamlined approach that simplifies the computations by using only the product of weight and activation magnitudes. Despite these advancements, these pruning methods share a common limitation: they uniformly set sparsity ratios for different layers in LLMs, failing to account for the varying importance of each layer in the model's overall performance. Intuitively, the front layers of LLMs are considered more important, as they establish the fundamental language understanding upon which the rest of the model relies. The limitations of uniform pruning contradict this intuition and lead to performance degradation of pruned LLMs with high sparsity ratios or on difficult language understanding tasks. However, achieving non-uniform pruning is also difficult, mainly due to two challenges: **(1) Additional costly computations:** Most non-uniform methods are trial-and-error paradigms requiring many computations and evaluation overheads. For example, BESA [58] performs differentiable iterative optimization for block-wise sparsity allocation based on evaluation results. However, this method primarily focuses on intra-layer sparsity configuration and takes at least 5 hours, which is considerably slower compared to other training-free approaches. Additionally, the overheads of these methods grow with both the number of layers and the sparse granularity of LLMs. **(2) Fixed and empirical allocations:** Recent methods like OWL [60] assign different sparsity ratios based on the outlier ratio within each layer. This empirical method requires tuning hyperparameters such as the outlier threshold and sparsity upper/lower bounds to obtain optimal values, and it heavily relies on empirical analysis and handcrafted design without providing a solid theoretical foundation for its effectiveness, making it difficult to scale to various models and datasets (see Figure 1 (***right***)). These dilemmas naturally raise the question: ***How can we efficiently obtain adaptive allocation strategies capable of handling different models and scenarios?***

To answer this question, we analyze the distribution of element-wise scores of different layers in LLMs based on existing sparse methods. As shown in Figure 1 (***left***), we observe that the mean values of the per-layer element importance scores of the front layers enjoy larger values. This observation aligns with OWL (see Figure 1 (***middle***)) and the understanding that the initial layers of LLMs are more important [13]. Furthermore, we find that other reduction operations, such as variance, entropy, etc., also yield distributions with similar trends. Motivated by these findings, we explore the possibility of directly utilizing these layer-wise importance scores as a guiding principle for allocating sparsity ratios across layers. Thus, we transform the reduction values of element-wise scores using various functions (e.g., sine, exponential) and employ the transformed values as layer-wise sparsity ratios. Such non-uniform manners bring promising gains, which are sensitive to specific transform operations. These observations inspire us to combine advanced reduction operations and transform functions to obtain an effective sparsity allocation based on the original importance scores from uniform pruning methods. However, how to obtain the most promising combinations? Fortunately, the recent advancements in AutoML [71, 45, 34] provide potential answers by enabling the automatic search for optimal solutions within a defined search space. By formulating this problem as an

AutoML task, we can leverage search algorithms to efficiently explore the search space and identify the effective combinations for non-uniform pruning in LLMs.

Based on our observations, we introduce DSA, an innovative framework that leverages expression discovery and evolutionary algorithms to tailor sparsity allocation schemes for LLMs. DSA seeks to find the best functions for mapping *element-wise scores* → *per-layer importances* → *sparsity ratios*. To identify the most effective allocation function, we construct a vast search space encompassing a diverse set of pre-process, reduction, transformation, and post-process operations. The pre-process operators, such as Frobenius norm and softmax, normalize the original importance values, enabling fair cross-layer comparison. Reduction operations, including variance, standard deviation, entropy, and geometric mean, extract the element-wise importance scores of each layer into a single representative value. Recognizing the potential for complex nonlinear relationships, we incorporate a wide range of mathematical functions as transformation operations, including sine and cosine. These functions provide flexibility to model intricate patterns and amplify or attenuate the importance scores as needed. Additionally, we introduce post-process operations to further increase the upper bound of the function fit. Within this rich search space, DSA employs an evolutionary algorithm to explore and discover promising allocation function candidates. The evolutionary process begins by initializing a population of diverse allocation functions, which are then iteratively evaluated and evolved through crossover and mutation operations. The crossover operation exchanges beneficial components between high-performing parent candidates, while mutation introduces random perturbations to promote exploration. Once the evolutionary process converges, DSA selects the top-performing allocation function candidates and seamlessly integrates them into existing pruning methods, such as Wanda and SparseGPT, through a plug-and-play mechanism. By leveraging the discovered allocation functions, these pruning methods can achieve significantly improved performance, maximizing compression while minimizing accuracy degradation. By automating the search process, DSA eliminates the need for manual tuning and expert intervention, reducing the time and effort required to find allocation strategies. The expressiveness of the search space and the ability to combine diverse operations enable the discovery of intricate, nonlinear allocation functions tailored to the unique characteristics of each LLM.

We conduct extensive experiments on publicly available language processing datasets and benchmarks. The experimental results demonstrate our method achieves significant performance gains on multiple challenging tasks such as arithmetic, knowledge reasoning, and multimodal tasks spanning GSM8K, MMLU, VQAv2, SQA, and VQA benchmarks across multiple model architectures including LLaMA-1|2|3, Mistral, Vicuna and OPT. Notably, our DSA method yields substantial improvements across all evaluated models, with peak gains of 14.58% in LLaMA-3 8B under magnitude pruning and 10.65% when integrated with SparseGPT. Even under high sparsity ratios of 60-70%, our method maintains robust performance, achieving improvements of 7.68% for LLaMA-2-13B at 60% sparsity. In multimodal tasks, DSA demonstrates exceptional capability by surpassing conventional pruning methods across all benchmarks, achieving superior scores of 76.08% on VQAv2, 65.57% on SQA, and 54.36% on VQA for LLaVA-1.5 with Vicuna-7B at 50% sparsity. The method's effectiveness is particularly evident in challenging scenarios, maintaining strong performance even under aggressive pruning conditions while consistently outperforming existing sparsity allocation approaches across model scales and architectures.

## 2   Related Work

Model compression techniques [24, 57], such as quantization [15, 40, 33] and sparsification [16, 65, 38], are practical approaches to reduce size of large language models (LLMs). Sparsification, or network pruning, increases the number of zero-valued weights and can be categorized into structured [39, 23] and unstructured [15, 66, 64] pruning. While determining pruning criteria and ratios is crucial, the massive scale of LLMs presents challenges for efficient pruning. Conventional retraining-based methods [65, 23] are often impractical due to high computational demands. Researchers develop LLM-specific pruning techniques that prioritize training-free and time-efficient approaches to address these challenges. SparseGPT [16] introduces an efficient Hessian matrix estimation technique to large-scale models. Wanda [49] further simplifies the approach, reducing the computational overhead and memory requirements. LLM-Pruner [39] examines model dependencies by incorporating first-order and approximated Hessian information, providing a comprehensive pruning approach. LLM Surgeon [53] adapts Kronecker-factored curvature approximations specifically

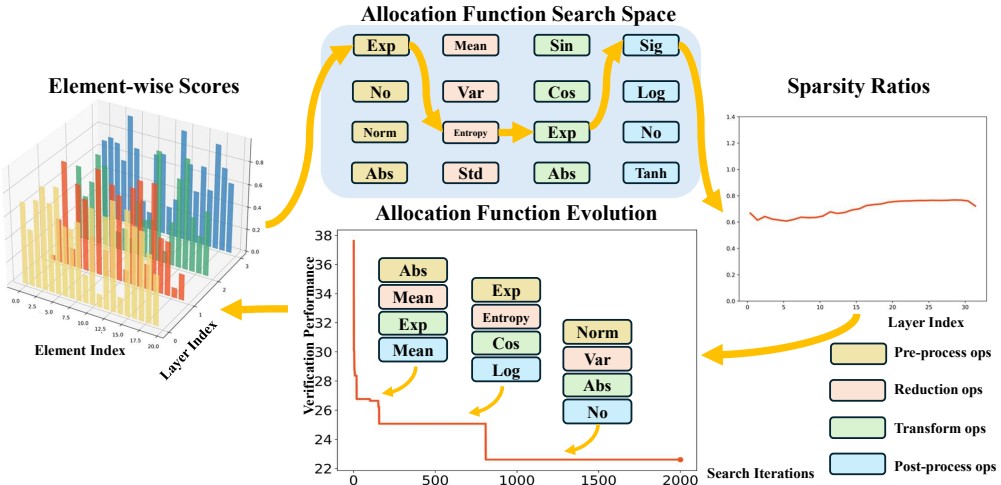

Figure 2: Overview of our DSA framework. We search for allocation functions to map element-wise scores to sparse ratios. We build pre-process, reduction, transform, and post-process operations as the search space for the allocation function, and then we perform evolutionary search.

for LLMs. Despite these advancements, most existing methods apply a uniform pruning rate across all layers, which may result in suboptimal performance. To address this, we present the first allocation function search for layer-adaptive sparsity, effectively minimizing performance degradation while achieving high compression ratios. Our method differs significantly from traditional layer-wise sparsity approaches for neural networks [12, 5, 26]. These methods often lead to the accumulation of errors across layers, as the pruning decisions for each layer are made independently without considering the global impact on the model's performance. Recent BESA [58] has shifted focus to intra-block sparsity allocation, employing various techniques to optimize the sparsity distribution within individual blocks or layers. FLAP [2] applies sparsity ratios process updating for performance compensation. In contrast, our method is layer-wise and training-free, with finer-grained allocation and an efficient process. OWL [15] requires experts' empirical design and tuning of hyperparameters. By automating the allocation process, our approach eliminates the need for manual intervention of OWL [15]. Our method differs from approaches like Pruner-Zero [44] in both the search object and the technique type. Pruner-Zero is a uniformly sparse method that employs a metric-optimized strategy. In contrast, our method explores non-uniform sparsity allocations, searching for an optimal allocation strategy. Additionally, our DSA method deviates from layer-dropping techniques [18, 14], which involve directly removing entire layers from the model architecture. Our approach preserves the model's overall architecture while strategically distributing sparsity across layers, allowing for high compression ratios without sacrificing significant performance or relying on extensive fine-tuning. More discussion are in Appendix A.

## 3 Methodology

### 3.1 Recap of Sparsity Methods for Large Language Models

Sparse methods introduce sparsity into the model weights by identifying and pruning redundant or less important weights for a given pre-trained dense weight matrix $\mathbf{W}$. To determine which weights should be pruned, sparse methods employ pruning metrics or importance scores, denoted as $\mathcal{S}(\mathbf{W}_l, \mathbf{X}_l, \mathbf{G}_l)$, where $\mathbf{W}_l$ represents the weights of layer $l$, and $\mathbf{X}_l$ and $\mathbf{G}_l$ are any layer-specific activations, gradient statistics, or the respective. The pruning metric ranks the weights based on their importance, and ranking results with the sparsity ratio $\phi$ serves as a threshold to select the most significant weights. The pruned weights are typically represented as a sparse mask $\mathbf{M}_l$, which is a binary tensor of the same shape as $\mathbf{W}_l$. The mask is obtained by applying a threshold function $f$ to the pruning metric $\mathcal{S}$ and the sparsity ratio $\phi_l$:

$$\mathbf{M}_l = f(\mathcal{S}(\mathbf{W}_l, \mathbf{X}_l, \mathbf{G}_l), \phi_l). \tag{1}$$

Table 1: Some operations in our search space. Full operations are in Appendix D.

| OP ID | OP Name | Expression | OP ID | OP Name | Expression |
|---|---|---|---|---|---|
| OP00 | Mean | $\mathrm{mean}(x)$ | OP09 | sigmoid | $\frac{1}{1+e^{-x}}$ |
| OP01 | std | $\sqrt{\mathrm{var}(x)}$ | OP10 | softmax | $\frac{e^x}{\sum_{i=1}^n e^{s_i}}$ |
| OP02 | var | $\mathrm{var}(x)$ | OP11 | exp | $e^x$ |
| OP03 | sqrt | $\sqrt{x}$ | OP12 | abslog | $|\ln x|$ |
| OP04 | geometric | $\sqrt[n]{\prod_{i=1}^n x_i}$ | OP13 | cosine | $cos(x)$ |
| OP05 | corref | $\frac{x^T x}{|x|_2^2}$ | OP14 | sine | $sin(x)$ |
| OP06 | l2_norm | $\frac{x-\mathrm{mean}(x)}{\mathrm{std}(x)}$ | OP15 | log | $lnx$ |
| OP07 | l1_norm | $|x|_1$ | OP16 | no_op | $x$ |
| OP08 | entropy | $-\sum_{i,j} x_{ij}\log x_{ij}$ | OP17 | rank | $\mathrm{rank}(x)$ |

The pruned weights $\mathbf{W}_l^{\mathrm{masked}}$ are then obtained by element-wise multiplication of the original weights $\mathbf{W}_l$ and the sparse mask $\mathbf{M}_l$:

$$\mathbf{W}_l^{\mathrm{masked}} = \mathbf{M}_l \odot \mathbf{W}_l, \tag{2}$$

where $\odot$ denotes element-wise multiplication. The choice of the pruning metric $\mathcal{S}$ and the sparsity ratio $\phi$ significantly impacts the effectiveness of the sparse method. The basic pruning metric is the magnitude-based approach [19], where $\mathcal{S}(\mathbf{W}_l) = |\mathbf{W}_l|$ employs the element-wise absolute value to assess weight significance. The sparsity ratio $\phi$ plays a crucial role in determining the level of sparsity introduced into the model. Higher values of $\phi$ correspond to higher levels of sparsity, resulting in greater reductions in model size and computational requirements. However, excessive pruning may lead to significant performance degradation if important weights are removed. In contrast to traditional methods that use a fixed sparsity ratio $\phi$ for all layers, our adaptive sparsity allocation scheme allows the sparsity ratio $\phi_l$ to vary across layers based on their importance. This approach is motivated by the observation that different layers in a deep neural network contribute differently to the overall model performance, and a uniform sparsity ratio may not be optimal. The advance of our adaptive sparsity allocation scheme is its ability to identify and selectively prune the less important layers, achieving better compression while preserving the model's performance.

## 4   Allocation Function Search Space

**Allocation Function Representation.** Our allocation function $\mathcal{A}$ aims to map element-wise scores $\mathcal{S}$ to per-layer importance values $\mathcal{V}$, and subsequently map these importance scores to sparsity ratios $\phi$. The allocation functions are represented as computation graphs consisting of various pre-process $\mathcal{T}_{\mathrm{pre}}$, reduction $\mathcal{T}_{\mathrm{reduce}}$, transformation $\mathcal{T}_{\mathrm{trans}}$, and post-process $\mathcal{T}_{\mathrm{post}}$ operations, as follows:

$$\mathcal{V} = \mathcal{T}_{\mathrm{reduce}}(\mathcal{T}_{\mathrm{pre}}(\mathcal{S})), \quad \phi = \mathcal{T}_{\mathrm{post}}(\mathcal{T}_{\mathrm{trans}}(\mathcal{V})), \quad \mathcal{A} = \mathcal{V}(\phi(\mathcal{S})). \tag{3}$$

**Motivation of Allocation Function Design.** As discussed in introduction and Figure 1 (left), our design is motivated by analyzing element-wise score distributions: (1) We notice that mean, variance, and entropy values of per-layer element-wise scores can serve as allocation indicators, inspiring reduction operations. (2) While basic reduction of element-wise scores showed modest improvements, applying transform functions yielded more promising results, prompting the introduction of transform operations. (3) We include pre-process to normalize scores for fair comparison and post-process to further enhance function fit's upper bound. These observations naturally encourage us to employ the four cascading operations for search space.

**Primary Operators.** Table 1 presents a subset of the primary operators considered in our search space, which is organized into four main categories:

- **Pre-process operations** $\mathcal{T}_{\mathbf{pre}}$ are applied to the element-wise scores $\mathcal{S}$ to prepare them for the subsequent reduction step. $\mathcal{T}_{\mathrm{pre}}$ standardizes inputs by normalizing scores across layers, ensuring consistent performance metrics by addressing scale variations. These operations can include clipping, normalization, or applying non-linear transformations.
- **Reduction operation** $\mathcal{T}_{\mathbf{reduce}}$ aggregates the pre-processed scores into a single per-layer importance score $\mathcal{V}$. $\mathcal{T}_{\mathrm{reduce}}$ condenses element-wise information by extracting representative values and reduces computational complexity. They use statistical measures like mean, standard deviation, variance, and entropy to provide insights into the distribution of the input data.

- **Transform operation** $\mathcal{T}_{\mathbf{trans}}$ models the distribution of per-layer scores $\mathcal{V}$ and transforms this into sparsity ratios $\phi$, enabling the representation of intricate patterns in layer importance. This can involve non-linear transformations like sigmoid, softmax, exponential, and logarithmic functions, which capture complex relationships, while trigonometric functions like sine and cosine capture periodic patterns or cyclical behaviors.
- **Post-process operation** $\mathcal{T}_{\mathbf{post}}$ plays the role of augmenting the fitting power and flexibility on transform operation. $\mathcal{T}_{\mathrm{post}}$ ensures that the sparsity ratios $\phi$ satisfy any required constraints, such as being between 0 and 1 across all layers. By combining these diverse operators, our framework constructs tailored allocation functions that capture the unique characteristics of each LLM.

## 5 Allocation Function Evolution

**Search Objectives.** Our search goal is to find the optimal combination of operations $\mathcal{T}$ that makes the sparse model perform optimally on the validation set given the sparsity metric $\mathcal{S}$ and the overall model size constraints $C$. This can be formulated as an optimization problem

$$\underset{\mathcal{T}}{\mathbf{argmax}} \quad \mathcal{P}(\mathbf{M} \odot \mathbf{W}, X, Y), \quad \text{s.t.} \quad \text{Size}(\mathbf{M} \odot \mathbf{W}) < C, \tag{4}$$

where $\mathbf{W}$ represents the weights of the LLMs, $X$ and $Y$ are the input and target data of the verification set, respectively, and $\mathcal{P}$ is the performance metric (*e.g.*, perplexity). The mask $\mathbf{M}$ is determined by the sparsity ratios $\phi$ through combination of operations $\mathcal{T}_{\mathrm{pre}}$, $\mathcal{T}_{\mathrm{reduce}}$, $\mathcal{T}_{\mathrm{trans}}$, and $\mathcal{T}_{\mathrm{post}}$, as

$$\mathbf{M} = f(\mathcal{S}, \phi) = f(\mathcal{S}, \mathcal{T}_{\mathrm{reduce}}(\mathcal{T}_{\mathrm{post}}(\mathcal{T}_{\mathrm{trans}}(\mathcal{T}_{\mathrm{pre}}(\mathcal{S}))))). \tag{5}$$

To solve this problem, we need to search a combination of $\mathcal{T}$ in 4 levels with around 10 options in each level, resulting in a rather large space (*i.e.*, $\mathcal{O}(10^4)$). In contrast to simple random search, we develop an evolutionary search for optimal allocation function.

**Evolution Search Procedure.** Our search process begins by generating an initial population of allocation function candidates, which can be created randomly or using heuristic techniques. Each candidate in this population corresponds to a unique combination of operations. Next, the performance of each candidate allocation function is evaluated. This involves computing the sparsity ratios by applying the candidate function to the sparsity metric, evaluating the pruned model on a validation set using a performance metric, and checking if the pruned model's size satisfies the given constraint. Based on this performance evaluation, the fittest candidates are selected for the next generation, considering criteria such as the performance metric, model size constraint, or a combination thereof. These selected candidates then undergo evolutionary operations like mutation and crossover to generate a new population of candidates for the subsequent iteration. The search process continues iterating until a stopping criterion is met, such as a maximum number of iterations or a satisfactory performance level. To accelerate the search, we employ various techniques: **(1) Program checking** uses static analysis to discard invalid candidates early, reducing computational overhead. **(2) Memoization and caching store** and reuse results from previous evaluations, avoiding redundant computations. **(3) Parallel evaluation** distributes the performance evaluation of different candidates across multiple computing resources. **(4) Surrogate models** approximate computationally expensive evaluations using techniques like neural networks trained on a subset of data. After each iteration, the performance of the best candidates is verified on a held-out validation set or a separate test set. These acceleration settings allow at least 100 times faster searches. In this way, we search our allocation function in **only 0.5 day on a 1× NVIDIA GPU H800 server** based on Wanda using perplexity results from the validation set of LLaMA-1-7B on WikiText2 [41]. We confirm that no search was performed on the test set, ensuring the comparisons are completely fair. In addition, the discovered allocation functions are transferable to other tasks without massive costs. Thus, the search cost can be spread across multiple pruning runs.

## 6 Discovered Allocation Function Analysis

One of the top-performing allocation functions discovered through the evolutionary search process is:

$$\mathcal{V}^* = \mathcal{T}^*_{\mathrm{reduce}}(\mathcal{T}^*_{\mathrm{pre}}(\mathcal{S})) = \sqrt[n]{\prod_{i=1}^{n} |\ln(\ln(\mathcal{S}))|_i}, \quad \phi^* = \mathcal{T}^*_{\mathrm{post}}(\mathcal{T}^*_{\mathrm{trans}}(\mathcal{V})) = \exp(\cos(\mathcal{V}^*)), \tag{6}$$

Table 2: Mean accuracies (%) of our DSA at 50% sparsity rate on 7 zero-shot tasks.

| Model | LLaMA-1 | | | | LLaMA-2 | | | LLaMA-3 | OPT |
| Method | 7B | 13B | 30B | 65B | 7B | 13B | 70B | 8B | 6.7B |
|---|---|---|---|---|---|---|---|---|---|
| Dense | 64.32 | 66.84 | 69.80 | 71.21 | 64.36 | 67.08 | 71.52 | 68.28 | 55.50 |
| Magnitude | 50.83 | 51.26 | 58.22 | 67.14 | 54.69 | 57.37 | 64.90 | 40.91 | 37.90 |
| **+ DSA (Ours)** | **53.73** | **59.05** | **60.55** | **67.86** | **57.90** | **61.38** | **68.76** | **55.50** | **40.25** |
| **Gain** | **2.90↑** | **7.78↑** | **2.33↑** | **0.72↑** | **3.21↑** | **4.01↑** | **3.85↑** | **14.58↑** | **2.35↑** |
| Wanda | 56.60 | 62.86 | 66.96 | 69.42 | 59.72 | 62.53 | 70.14 | 55.93 | 45.19 |
| **+ DSA (Ours)** | **59.22** | **63.03** | **67.80** | **70.98** | **60.80** | **64.87** | **70.54** | **60.70** | **45.45** |
| **Gain** | **2.62↑** | **0.17↑** | **0.84↑** | **1.56↑** | **1.09↑** | **2.34↑** | **0.40↑** | **4.76↑** | **0.26↑** |
| SparseGPT | 53.60 | 62.08 | 63.97 | 67.20 | 54.26 | 57.92 | 68.02 | 51.77 | 52.38 |
| **+ DSA (Ours)** | **58.33** | **62.49** | **67.63** | **67.32** | **59.22** | **64.10** | **68.70** | **62.41** | **55.15** |
| **Gain** | **4.73↑** | **0.41↑** | **3.66↑** | **0.12↑** | **4.95↑** | **6.18↑** | **0.68↑** | **10.65↑** | **2.77↑** |

where $\mathcal{T}_{\text{pre}}^*$ consists of two steps: log and abslog. The log step applies the natural logarithm operation $\ln(\mathcal{S})$ to the input importance values $\mathcal{S}$, compressing the range of values and potentially highlighting differences in smaller values. The abslog step computes the absolute value of the natural logarithm, $|\ln(\mathcal{S})|$, ensuring that negative values are treated symmetrically with positive values, preventing potential cancellations or sign changes. $\mathcal{T}_{\text{reduce}}^*$ applies the geometric mean operator $\sqrt[n]{\prod_{i=1}^{n} x_i}$ to the result of abslog. This operation further compresses the range of values and introduces a nonlinear transformation. $\mathcal{T}_{\text{trans}}^*$ is the cosine function $\cos(x)$, applied to the output of the geometric mean. This periodic function introduces oscillatory behavior, which can capture potential cyclical patterns or dependencies in the importance values. Finally, $\mathcal{T}_{\text{post}}^*$ applys the exponential function $\exp(x)$ to the result of the cosine operation. This step reintroduces nonlinearity and expands the range of values, potentially amplifying or attenuating the importance scores as needed.

**Stability Analyses.** To show that the function $\mathcal{V} = \sqrt[n]{\prod_{i=1}^{n} |\ln(\ln(\mathcal{S}))|_i}$ is stable under small perturbations in the input $\mathcal{S}$, we can derive an expression for the difference $\mathcal{V}(\mathcal{S} + \Delta\mathcal{S}) - \mathcal{V}(\mathcal{S})$ and analyze its behavior for small $\Delta\mathcal{S}$. The difference is approximately:

$$\mathcal{V}(\mathcal{S} + \Delta\mathcal{S}) - \mathcal{V}(\mathcal{S}) \approx \frac{1}{n}\mathcal{V}(\mathcal{S})^{1-n}\left(\sum_{i=1}^{n} \text{sgn}(\ln(\ln(\mathcal{S})))\frac{1}{\ln(\mathcal{S})}\frac{1}{\mathcal{S}}\prod_{j\neq i} k_j(\mathcal{S})\right)\Delta\mathcal{S} + \mathcal{O}((\Delta\mathcal{S})^2). \quad (7)$$

For small $\Delta\mathcal{S}$, the second-order term $\mathcal{O}((\Delta\mathcal{S})^2)$ becomes negligible, and the leading term is linear in $\Delta\mathcal{S}$. The coefficient of $\Delta\mathcal{S}$ in this leading term is a product of bounded functions of $\mathcal{S}$. Therefore, for small perturbations $\Delta\mathcal{S}$ around any positive value of $\mathcal{S}$, the difference is also small, and the function $\mathcal{V}(\mathcal{S})$ is stable under such perturbations. More analyses are in the Appendix B.

# 7 Experiments

In this section, we conduct detailed evaluation experiments on multiple tasks and models. For pruning and evaluation, we follow the settings of Wanda, SparseGPT and ensure using the same database version, GPU model, and random seed across all experiments to maintain consistent conditions. More experimental results are in Appendix C.

## 7.1 Experiments on Zero-shot Tasks

**Implementation.** To verify the effectiveness and generalizability, we perform extensive evaluation of our models on 7 zero-shot tasks. We employ a set of seven tasks sourced from the EleutherAI LM Harness [50]. These tasks include Winogrande [46], OpenBookQA [42], HellaSwag [62], BoolQ [6], ARC [7], and RTE [54]. To assess the performance of our Dynamic Sparse Allocation (DSA) method, we evaluate its effectiveness on several models. These LLMs include LLaMA-1 (7B/13B/30B/65B) [51], LLaMA-2 (7B/13B/70B) [52], LLaMA-3 (8B) [1], and OPT (6.7B/13B) [63]. Our allocation function is applied to different pruning methods, namely Wanda [49], Magnitude-based pruning [20], and SparseGPT [15]. For fair comparisons, we follow the same configurations of SparseGPT and Wanda methods. We select data from the C4 dataset and ensure that all test data used in the evaluation are from zero-shot settings.

Table 3: Mean accuracies (%) of our DSA on 7 zero-shot tasks at 60% sparsity rates.

| Method | LLaMA-2-7B | LLaMA-2-13B | LLaMA-3-70B |
|---|---|---|---|
| Magnitude | 50.81 | 51.16 | 55.86 |
| + DSA (Ours) | 57.84 | 54.28 | 60.24 |
| Gain | 7.03↑ | 3.12↑ | 4.38↑ |
| Wanda | 60.90 | 72.00 | 40.51 |
| + DSA (Ours) | 62.08 | 73.24 | 42.74 |
| Gain | 1.18↑ | 1.24↑ | 2.23↑ |
| SparseGPT | 60.68 | 70.14 | 65.03 |
| + DSA (Ours) | 61.31 | 72.12 | 67.34 |
| Gain | 0.63↑ | 1.98↑ | 2.31↑ |

Table 4: Mean accuracies (%) of our DSA on 7 zero-shot tasks at 70% sparsity rates.

| Method | LLaMA-2-7B | LLaMA-2-13B | LLaMA-3-70B |
|---|---|---|---|
| Magnitude | 35.61 | 38.38 | 38.76 |
| + DSA (Ours) | 37.95 | 46.06 | 42.98 |
| Gain | 2.34↑ | 7.68↑ | 4.22↑ |
| Wanda | 36.08 | 41.46 | 40.44 |
| + DSA (Ours) | 38.00 | 43.18 | 42.78 |
| Gain | 1.92↑ | 1.72↑ | 2.34↑ |
| SparseGPT | 43.61 | 48.76 | 43.22 |
| + DSA (Ours) | 45.56 | 50.04 | 45.73 |
| Gain | 1.95↑ | 1.28↑ | 2.51↑ |

**Sparsity Results on Varying Models.** The results in Table 2 demonstrate the effectiveness of our allocation function in improving the performance of pruned models across various methods and model architectures. When integrated with magnitude-based pruning, DSA yields substantial improvements across all evaluated models, with particularly impressive gains in LLaMA-3 8B, where accuracy increases by 14.58%. For the Wanda pruning method, DSA consistently enhances performance, achieving notable improvements of 4.76% in LLaMA-3 8B and 2.62% in LLaMA-1 7B, while maintaining stable gains across larger models such as LLaMA-2 70B with a 0.40% increase. The integration of DSA with SparseGPT produces the most striking results, with substantial improvements of 10.65% for LLaMA-3 8B and 6.18% for LLaMA-2 13B, demonstrating its exceptional capability to optimize sparsity patterns. DSA shows particular strength in enhancing smaller models, with LLaMA-1 7B experiencing gains of 4.73% under SparseGPT and 2.90% under magnitude pruning, while also maintaining effectiveness across larger architectures such as LLaMA-2 70B, where it achieves improvements of 3.85% under magnitude pruning. These consistent performance improvements across different model scales, from the 6.7B OPT to the 70B LLaMA-2, highlight the ability of DSA to migrate well and generalize across different pruning techniques and model architectures, enabling improved performance and efficient compression of LLMs while minimizing the impact on their zero-shot capabilities.

**Sparsity Results under High Pruning Ratios.** The experimental results in Table 3 and Table 4 demonstrate the robust performance of DSA under high pruning ratios across different model scales and pruning methods. When integrated with magnitude-based pruning, DSA exhibits remarkable improvements, achieving gains of up to 7.68% for LLaMA-2-13B at 60% sparsity and 7.03% for LLaMA-2-7B at 70% sparsity. In combination with Wanda, DSA consistently enhances performance across all models and sparsity ratios, with particularly notable improvements in LLaMA-3-70B, where it achieves gains of 2.23% and 2.34% at 60% and 70% sparsity respectively. The integration with SparseGPT yields steady improvements, with the most significant gains observed in LLaMA-3-70B (2.51% at 70% sparsity) and LLaMA-2-7B (1.95% at 60% sparsity). DSA's effectiveness is particularly evident in challenging scenarios, such as maintaining LLaMA-2-13B's performance at 73.24% accuracy even under 70% sparsity when combined with Wanda, and achieving 67.34% accuracy with LLaMA-3-70B at 60% sparsity when integrated with SparseGPT, demonstrating its capability to preserve model performance even under aggressive pruning conditions.

**Compare Other Sparsity Allocation Methods.** Table 5 shows WikiText-2 perplexity results demonstrate the superior performance of DSA across varying high sparsity rates from 65% to 80% in LLaMA-1-7B. At 65% sparsity, DSA achieves the lowest perplexity of 12.62, outperforming OWL's 13.05 and showing substantial improvement over traditional methods like Uniform (20.85) and ER (45.85). The performance advantage of DSA becomes more pronounced as sparsity increases, reaching a perplexity of 736.81 at 80% sparsity, which represents a significant improvement over OWL (1002.87) and BESA (2208.75). Notably, DSA demonstrates remarkable stability under extreme sparsification, maintaining performance far superior to conventional approaches like Global and ER-plus, which deteriorate dramatically with perplexities of 39918.56 and 6013.91 respectively at 80% sparsity. The consistent superiority of DSA across all sparsity levels, particularly its ability to maintain relatively low perplexity even at 80% sparsity, validates its effectiveness in allocating sparsity while preserving model performance. By constructing tailored allocation functions that capture the unique characteristics of each layer, DSA achieves superior performance compared to other commonly used layerwise sparsity methods, especially at higher sparsity rates.

Table 5: WikiText-2 perplexity ($\downarrow$) performance of various allocation methods with the Wanda metric for sparse LLaMA-1-7B at varying high sparsity rates (65%~80%).

| Method | Global [60] | ER-plus [36] | ER [43] | Uniform [69] | BESA [58] | OWL [60] | DSA (Ours) |
|---|---|---|---|---|---|---|---|
| 65% | 867.82 | 97.28 | 45.85 | 20.85 | 18.52 | 13.05 | **12.62** |
| 70% | 5147 | 229.17 | 112.03 | 81.18 | 42.58 | 24.54 | **22.60** |
| 75% | 25863.75 | 1482.93 | 3287.92 | 927.42 | 257.89 | 152.47 | **103.32** |
| 80% | 39918.56 | 6013.91 | 11151.18 | 3499.88 | 2208.75 | 1002.87 | **736.81** |

Table 6: 50% Sparsity results (%) on GSM8K.

| Method | LLaMA-1 | | LLaMA-2 | | Mistral |
| | 7B | 13B | 7B | 13B | 7B |
|---|---|---|---|---|---|
| Dense | 11.07 | 17.82 | 14.59 | 19.86 | 40.11 |
| Magnitude | 1.52 | 5.99 | 2.05 | 6.22 | 15.53 |
| SparseGPT | 8.19 | 15.60 | 8.11 | 13.42 | 25.40 |
| Wanda | 7.96 | 11.52 | 7.43 | 9.10 | 22.74 |
| **Ours** | **8.22** | **15.64** | **8.47** | **14.27** | **25.78** |

Table 7: 50% Sparsity results (%) on MMLU.

| Method | LLaMA-1 | | LLaMA-2 | | Mistral |
| | 7B | 13B | 7B | 13B | 7B |
|---|---|---|---|---|---|
| Dense | 35.28 | 46.98 | 41.97 | 51.47 | 58.92 |
| Magnitude | 26.24 | 30.12 | 26.04 | 43.83 | 50.83 |
| SparseGPT | 29.48 | 38.29 | 33.06 | 47.14 | 50.95 |
| Wanda | 29.81 | 37.84 | 32.09 | 48.06 | 53.05 |
| **Ours** | **31.05** | **39.76** | **33.08** | **48.38** | **53.87** |

Table 8: Results (%) on 7B LLaVA-1.5.

| Vicuna-7B | VQAv2 | SQA | VQA |
|---|---|---|---|
| Dense | 78.50 | 66.80 | 58.20 |
| Magnitude (50%) | 63.50 | 31.24 | 38.39 |
| SparseGPT (50%) | 75.86 | 63.92 | 53.69 |
| Wanda (50%) | 75.72 | 63.99 | 53.05 |
| **Ours (50%)** | **76.08** | **65.57** | **54.36** |
| Wanda (4:8) | 72.70 | 58.92 | 50.20 |
| **Ours (4:8)** | **73.54** | **59.84** | **51.74** |
| Wanda (2:4) | 68.92 | 55.06 | 45.42 |
| **Ours (2:4)** | **71.18** | **57.44** | **48.25** |

Table 9: Results (%) on 13B LLaVA-1.5.

| Vicuna-13B | VQAv2 | SQA | VQA |
|---|---|---|---|
| Dense | 80.00 | 74.94 | 61.30 |
| Magnitude (50%) | 75.79 | 70.95 | 52.16 |
| SparseGPT (50%) | 78.62 | 71.19 | 58.23 |
| Wanda (50%) | 78.58 | 70.97 | 58.03 |
| **Ours (50%)** | **79.10** | **73.17** | **58.70** |
| Wanda (4:8) | 77.57 | 69.79 | 56.15 |
| **Ours (4:8)** | **77.79** | **70.50** | **56.33** |
| Wanda (2:4) | 75.39 | 64.89 | 52.52 |
| **Ours (2:4)** | **76.75** | **67.13** | **54.05** |

## 7.2 Experiments on Arithmetric & Knowledge Reasoning Tasks

**Implementation.** We apply our allocation function to Wanda and evaluate the performance on arithmetic and knowledge reasoning tasks, specifically on the GSM8K [8] and MMLU [22] datasets using LLaMA-1 7B/13B, LLaMA-2 7B/13B, and Mistral 7B models [25].

**Comparison Results.** On GSM8K (Table 6), our method consistently outperforms baselines like magnitude pruning, SparseGPT and Wanda across all evaluated LLaMA-1, LLaMA-2, and Mistral models. The gains are most notable for smaller models like LLaMA-1 7B. Similarly, on MMLU (Table 7), our DSA achieves the highest accuracy among all methods, outperforming Wanda by up to 1.24% on LLaMA-1 7B and showing consistent improvements across larger LLaMA and Mistral models. The results highlight the effectiveness of our allocation strategy in optimizing sparse patterns across architectures, even on challenging reasoning tasks.

## 7.3 Experiments on Multimodal Tasks

**Implementation.** To explore the applicability of our method towards a more diverse task, we evaluate our method for pruning language models on various visual question-answering and reasoning benchmarks, including VQAv2 [17], SQA [37], and VQA [47]. In particular, our method is applied with Wanda to LLaVA-1.5 [35], where the Vicuna-7B and Vicuna-13B language models are pruned. In addition, we also transfer some pruning methods and make comparisons on these multimodal tasks.

**Comparison Results.** Table 8 and Table 9 showcase the performance of different pruning methods such as Magnitude, SparseGPT, and Wanda on the Vicuna-7B and Vicuna-13B models. For LLaVA-1.5 with Vicuna-7B, at 50% sparsity, our method surpasses conventional pruning methods across all benchmarks, achieving top scores of 76.08% on VQAv2, 65.57% on SQA, and 54.36% on VQA. Under the 4:8 structured sparsity pattern, our method consistently outperforms Wanda across all metrics, showing improvements of 0.84%, 0.92%, and 1.54% on VQAv2, SQA, and VQA respectively for the 7B model. The performance gap becomes even more pronounced with 2:4 sparsity, where

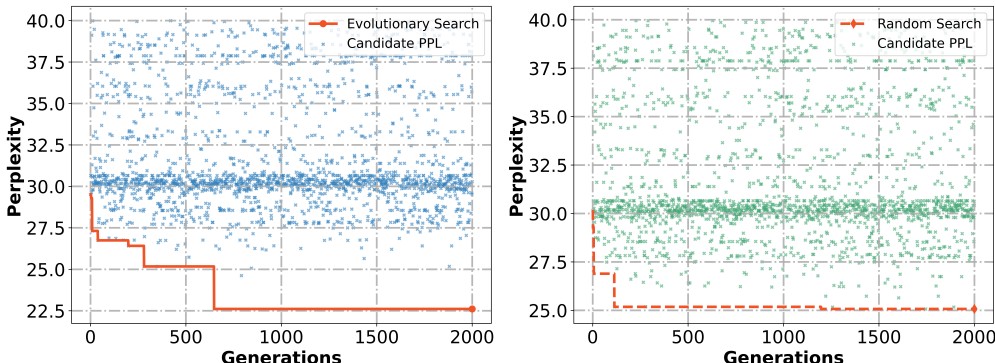

Figure 3: Comparison of search curves of evolution search and random search in our sparse allocation function discovery for LLaMA-1 7B on WikiText-2. Evolutionary search converges faster than random search and can achieve potential results with better perplexity (↓) performance.

our method achieves substantial gains of 2.26%, 2.38%, and 2.83% over Wanda on the same metrics. For the 13B model, while the improvements under 4:8 sparsity are modest (0.22%, 0.71%, and 0.18%), the 2:4 pattern shows more significant gains of 1.36%, 2.24%, and 1.53% on VQAv2, SQA, and VQA respectively. The consistent superiority of our method across different model sizes and sparsity patterns demonstrates its robustness and effectiveness in maintaining model performance under aggressive compression settings.

### 7.4 Analysis

**Search Algorithm Analysis.** Figure 3 compares random search with our evolutionary search in the function search tasks. Our advanced evolutionary search has faster convergence and final results, *e.g.*, our search algorithm exceeds in 700 generations over 1500 generations of the random algorithm.

**Sparse Allocation Results Analysis.** Figure 1 illustrates the per-layer importance values and the final sparsity ratios with our allocation functions. These distributions are nicely tailored to the specific and explanatory nature of the LLMs.

## 8 Conclusion

In this paper, we introduce the DSA framework, which offers a powerful and automated approach to discovering tailored sparsity allocation schemes for LLMs. By leveraging expression discovery and evolutionary algorithms, DSA can effectively explore a vast search space of operations and uncover intricate, nonlinear allocation functions that map importance metrics to optimal layer-wise sparsity ratios. This automated process eliminates manual tuning and expert intervention, reducing the time and effort required for effective sparsity allocation. Our DSA demonstrates promising results on the LLaMA, Mistral, and OPT models. We hope the DSA framework and its underlying principles will provide valuable insights to the research community, inspiring new avenues for efficient and effective compression of LLMs and enabling their wider deployment in resource-constrained environments.

**Limitations.** Following the AutoML technical route [29, 11, 28], we also need some cost in search process. We will develop more efficient search algorithms and incorporate domain knowledge to guide and constrain the search process in future work.

## Acknowledgements

The research was supported by Theme-based Research Scheme (T45-205/21-N) from Hong Kong RGC, Hong Kong CRF grants under Grant No. C7004-22G and Generative AI Research and Development Centre from InnoHK.

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

# Appendix

# A  More Discussion

## A.1  Comparison with Existing Methods

**Compare to existing AutoML techniques.** In contrast to common AutoML [10, 70, 68] and evolutionary algorithms [31, 30, 48, 27], our method introduces several groundbreaking innovations specifically tailored for sparsity allocation discovery in LLMs. We are the first to frame LLM sparsity allocation as an AutoML problem, opening new avenues for optimizing LLM efficiency. Our approach introduces a distinctive search space customized for LLM sparsity allocation, combining pre-processing, reduction, transformation, and post-processing operations in novel ways, allowing for more nuanced and effective sparsity distributions. Diverging from typical AutoML methods such as NAS and HPO that search for specific modes or hyperparameters, our framework emphasizes generalized function discovery, identifying common patterns across LLMs and formulating interpretable sparsity allocation. Furthermore, we develop LLM-specific acceleration techniques to reduce search time, making our DSA practical for large-scale LLM optimization.

**Compare to ECoFLaP [59].** Our work represents the first automated search for adaptive sparsity methods, which significantly differs from traditional adaptive pruning methods like ECoFLaP. We employ an automated search method that eliminates the need for expert design and adapts strategies to different models and tasks, whereas ECoFLaP relies on hand-designed, hyperparameter tuning. Our comprehensive search space systematically maps element-wise scores to per-layer importances to sparsity ratios, in contrast to ECoFLaP's simple linear computation of keep ratio during its two-stage pruning. Notably, our method obtains significant performance gains across various large language and multimodal models, demonstrating superior performance compared to ECoFLaP.

**Compare to OWL.** When compared to OWL [15], our method revolutionizes the field with an automated search approach that removes the need for expert design, while OWL remains constrained by hand-designed hyperparameter tuning and fixed form constraints. Our comprehensive search space systematically maps element-wise scores to sparsity ratios, surpassing OWL's limited linear computation of keep ratio based on outlier ratio. Comparative experiments definitively demonstrate our method's significant outperformance over OWL, as detailed in Table 8.

**Compare to Pruner-Zero** Our DSA approach differs fundamentally from Pruner-Zero [44] across multiple dimensions including method type, search space, task, strategy, and Input-Output characteristics. We uniquely frame LLM sparsity allocation as an AutoML challenge, opening novel avenues for enhancing LLM efficiency. Our search space is specifically customized for LLM sparsity allocation, integrating various operations in innovative ways. Additionally, we develop LLM-specific acceleration techniques like program checking, making our approach practical for large-scale LLM optimization.

**Compare to existing ayer-wise sparsity approaches.** Our method differs significantly from traditional layer-wise sparsity approaches for neural networks [12, 5, 26, 32]. These methods often lead to the accumulation of errors across layers, as the pruning decisions for each layer are made independently without considering the global impact on the model's performance. Furthermore, the extensive retraining required on vast datasets further amplifies the challenges associated with applying these techniques to LLMs. In contrast to our approach, recent work such as BESA [58] has shifted focus to intra-block sparsity allocation, employing various techniques to optimize the sparsity distribution within individual blocks or layers. Despite operating at a finer granularity, these methods fundamentally adhere to a layer-wise pruning paradigm, neglecting the importance of global sparsity allocation across the entire model. Consequently, the resulting allocation may be locally optimal within each layer but globally suboptimal, potentially leading to solutions that are stuck in local optima and fail to fully leverage the potential benefits of non-uniform sparsity distribution. Our DSA method addresses these limitations by introducing a holistic approach that considers the global impact of sparsity allocation across all layers. Unlike traditional layer-wise pruning methods that operate independently on each layer, our method employs an efficient search process to discover an allocation function that dynamically determines the appropriate sparsity level for each layer based on its characteristics and contribution to the overall model performance.

Table 10: Comparison of Method Characteristics of our DSA and Pruner-Zero.

| Method | Types | Task | Search space | Input | Output | Strategy |
|--------|-------|------|--------------|-------|--------|----------|
| Pruner-Zero | uniform | symbolic Pruning Metric | unary/binary operations | element-wise weights/gradients | element-wise score | Symbolic Regression |
| DSA (ours) | non-uniform | adaptive allocation function | pre-process/reduction/ transform/post-process | element-wise score | layer-wise sparsity ratios | Evolutionary Algorithm |

## B  Theoretical Understanding of Discovered Allocation Function

To show that the function $\mathcal{V} = \sqrt[n]{\prod_{i=1}^{n} |\ln(\ln(\mathcal{S}))|_i}$ is stable under small perturbations in the input $\mathcal{S}$, we can derive an expression for the difference $\mathcal{V}(\mathcal{S} + \Delta\mathcal{S}) - \mathcal{V}(\mathcal{S})$ and analyze its behavior for small $\Delta\mathcal{S}$. We define the following intermediate functions:

$$g(\mathcal{S}) = \ln(\mathcal{S})$$
$$h_i(\mathcal{S}) = \ln(g(\mathcal{S})) = \ln(\ln(\mathcal{S}))$$
$$k_i(\mathcal{S}) = |h_i(\mathcal{S})| = |\ln(\ln(\mathcal{S}))|_i$$
$$l(\mathcal{S}) = \prod_{i=1}^{n} k_i(\mathcal{S})$$
$$\mathcal{V}(\mathcal{S}) = \sqrt[n]{l(\mathcal{S})} = \sqrt[n]{\prod_{i=1}^{n} |\ln(\ln(\mathcal{S}))|_i}$$

Using the chain rule for differentiation, we can derive the derivative of $\mathcal{V}(\mathcal{S})$ with respect to $\mathcal{S}$ as follows:

$$\frac{d\mathcal{V}}{d\mathcal{S}} = \frac{1}{n} \sqrt[n]{l(\mathcal{S})}^{1-n} \frac{dl}{d\mathcal{S}}$$
$$= \frac{1}{n} \mathcal{V}(\mathcal{S})^{1-n} \left( \sum_{i=1}^{n} \frac{dk_i}{d\mathcal{S}} \prod_{j \neq i} k_j(\mathcal{S}) \right)$$
$$= \frac{1}{n} \mathcal{V}(\mathcal{S})^{1-n} \left( \sum_{i=1}^{n} \operatorname{sgn}(h_i(\mathcal{S})) \frac{dh_i}{d\mathcal{S}} \prod_{j \neq i} k_j(\mathcal{S}) \right)$$
$$= \frac{1}{n} \mathcal{V}(\mathcal{S})^{1-n} \left( \sum_{i=1}^{n} \operatorname{sgn}(\ln(\ln(\mathcal{S}))) \frac{1}{\ln(\mathcal{S})} \frac{1}{\mathcal{S}} \prod_{j \neq i} k_j(\mathcal{S}) \right)$$

where $\operatorname{sgn}(\ln(\ln(\mathcal{S})))$ is the sign function, equal to 1 for $\ln(\ln(\mathcal{S})) > 0$ and -1 for $\ln(\ln(\mathcal{S})) < 0$.

Now, we can use Taylor's theorem to approximate $\mathcal{V}(\mathcal{S} + \Delta\mathcal{S})$ around $\mathcal{S}$ as:

$$\mathcal{V}(\mathcal{S} + \Delta\mathcal{S}) \approx \mathcal{V}(\mathcal{S}) + \frac{d\mathcal{V}}{d\mathcal{S}}(\mathcal{S})\Delta\mathcal{S} + \mathcal{O}((\Delta\mathcal{S})^2)$$
$$= \mathcal{V}(\mathcal{S}) + \frac{1}{n} \mathcal{V}(\mathcal{S})^{1-n} \left( \sum_{i=1}^{n} \operatorname{sgn}(\ln(\ln(\mathcal{S}))) \frac{1}{\ln(\mathcal{S})} \frac{1}{\mathcal{S}} \prod_{j \neq i} k_j(\mathcal{S}) \right) \Delta\mathcal{S} + \mathcal{O}((\Delta\mathcal{S})^2)$$

Therefore, the difference $\mathcal{V}(\mathcal{S} + \Delta\mathcal{S}) - \mathcal{V}(\mathcal{S})$ is approximately:

$$\mathcal{V}(\mathcal{S} + \Delta\mathcal{S}) - \mathcal{V}(\mathcal{S}) \approx \frac{1}{n} \mathcal{V}(\mathcal{S})^{1-n} \left( \sum_{i=1}^{n} \operatorname{sgn}(\ln(\ln(\mathcal{S}))) \frac{1}{\ln(\mathcal{S})} \frac{1}{\mathcal{S}} \prod_{j \neq i} k_j(\mathcal{S}) \right) \Delta\mathcal{S} + \mathcal{O}((\Delta\mathcal{S})^2)$$

**Algorithm 1** Evolutionary Search for Allocation Function Discovery

---
**Input**: Search space $\mathcal{S}$, population size $P$, max iterations $T$, sample ratio $r$, top-k $k$.
**Output**: Best allocation function $\mathcal{A}^*$.

1: Initialize population $\mathcal{P}0$ with $P$ random allocation functions;
2: **for** $i = 1, 2, \ldots, T$ **do**
3:      Sample pool $\mathcal{R} \subset \mathcal{P}i - 1$ with ratio $r$;
4:      Top candidates $G_{ik} :=$ GetTopk($\mathcal{R}$, $k$);
5:      Parent $G_i^p :=$ RandomSelect($G_{ik}$);
6:      Mutant $G_i^m :=$ MUTATE($G_i^p$);
7:      Crossover $G_i^c :=$ CROSSOVER($G_i^p$, RandomSelect($G_{ik} \setminus G_i^p$));
8:      $\mathcal{P}i := \mathcal{P}i - 1 \cup G_i^m, G_i^c$;
9:      $\mathcal{P}_i :=$ Select-Top-Performers($\mathcal{P}i$, $P$);
10: **end for**
11: $\mathcal{A}^* :=$ argmaxA $\in \mathcal{P}_{\text{PPL}}(\mathcal{A}(\mathcal{S}), \phi)$;

---

For small $\Delta\mathcal{S}$, the second-order term $\mathcal{O}((\Delta\mathcal{S})^2)$ becomes negligible, and the leading term is linear in $\Delta\mathcal{S}$. The coefficient of $\Delta\mathcal{S}$ in this leading term is a product of bounded functions of $\mathcal{S}$, namely:

- $\mathcal{V}(\mathcal{S})^{1-n}$, which is bounded for $\mathcal{S} > 0$ - $\text{sgn}(\ln(\ln(\mathcal{S})))$, which is either 1 or -1 - $\frac{1}{\ln(\mathcal{S})}$, which is bounded for $\mathcal{S} > 1$ - $\frac{1}{\mathcal{S}}$, which is bounded for $\mathcal{S} > 0$ - $\prod_{j \neq i} k_j(\mathcal{S})$, which is bounded for finite $\mathcal{S}$

Therefore, for small perturbations $\Delta\mathcal{S}$ around any positive value of $\mathcal{S}$, the difference $\mathcal{V}(\mathcal{S} + \Delta\mathcal{S}) - \mathcal{V}(\mathcal{S})$ is also small, and the function $\mathcal{V}(\mathcal{S})$ is stable under such perturbations.

This stability property is desirable for the function $\mathcal{V}(\mathcal{S})$, as it ensures that small fluctuations or measurement errors in the input importance values $\mathcal{S}$ do not significantly impact the computed result $\mathcal{V}$, leading to robust and consistent computations.

## C More Experiment Details

Table 11: WikiText-2 perplexity ($\downarrow$) performance of our searched allocation function with different initial seeds with the Wanda metric for 70% sparse LLaMA-1-8B.

| Seed | Searched Allocation Functions | perplexity |
|---|---|---|
| Seed-1 | ABSLOG-VAR-ATAN,ASIN | 24.52 |
| Seed-2 | ABSLOG,NO_OP-VAR-ATAN,ACOS | 23.69 |
| Seed-3 | LOG,ABSLOG-GEOMETRIC_MEAN-ACOS,SIGMOID | 22.60 |
| Seed-4 | LOG,ABSLOG-GEOMETRIC_MEAN-COS,EXP | 22.60 |
| Seed-5 | MMS,ABSLOG-VAR-ATAN,ASIN | 24.61 |

### C.1 More details about Evolutionary Search

We commence our experiments by searching for an optimal allocation function based on the Wanda pruning method, utilizing perplexity results from the validation set of the LLaMA-1-7B model on the WikiText2 dataset [41]. We ensure that no search is performed on the test set, maintaining a fair and unbiased comparison. Subsequently, we directly transfer this discovered allocation function to different tasks and scenarios without conducting additional searches, aiming to evaluate its generalizability. To establish a robust and reliable experimental setup, we allocate 20% of the original dataset's training set as a held-out test set for the search process. We meticulously confirm that these validation datasets do not overlap with the test set, preventing any potential data leakage or bias in our evaluations. During the search phase, we configure the evolutionary algorithm (Algorithm 1) with a population size of 20, a maximum of 1,000 iterations, a sample ratio of 0.9, and a top-k value of 5. Throughout this process, we evaluate a total of 50 allocation function candidates within the validation set, iteratively refining and improving the solutions through the evolutionary mechanisms of crossover and mutation. By adhering to this rigorous experimental protocol, we ensure the integrity and validity of our results, enabling a comprehensive assessment of the discovered allocation function's effectiveness and its ability to generalize across diverse tasks and datasets.

**C.2  Analysis of Search Robustness**

Our evolutionary search algorithm shows robustness to different initialization seeds both theoretically and experimentally. Theoretically, it maintains robustness by: (1) Starting with a diverse initial population of allocation functions to avoid getting trapped in poor solutions. (2) Using genetic operators like mutation and crossover to explore new regions beyond the initial population. Experimentally, we evaluated the algorithm across 5 different random initialization seeds when searching for an allocation function to sparsify LLaMA-1-8B on WikiText-2 to 70% sparsity using the Wanda metric. The results in Table 11 show: (1) Different seeds discovered different allocation function expressions involving various operations. Despite this variation, all seeds achieved competitive perplexity performance in the range of 22.60 - 24.61. (2) Two seeds (3 and 4) converged to the same best perplexity of 22.60 despite different initial conditions. This consistent performance across diverse initializations demonstrates the robustness of the search aided by the diverse initial population and exploration via genetic operators. While minor performance variations exist, the overall competitiveness of the results validates the algorithm's resilience against initialization biases through effective search space exploration.

# D  Search Space for Allocation Functions

The search space for allocation functions is organized into four main categories: (1) pre-process operations $\mathcal{T}_{\text{pre}}$, (2) reduction operations $\mathcal{T}_{\text{reduce}}$, (3) transform operations $\mathcal{T}_{\text{trans}}$, and (4) post-process operations $\mathcal{T}_{\text{post}}$. The following subsections provide a detailed list of the operations included in each category, along with their corresponding mathematical formulas.

## D.1  Pre-process Operations $\mathcal{T}_{\text{pre}}$

The pre-process operations $\mathcal{T}_{\text{pre}}$ are applied to the element-wise scores $\mathcal{S}$ to prepare them for the subsequent reduction step. The operations in this category include:

- **NO_OP**: No operation is performed, and the input is returned as is.
$$\text{NO\_OP}(x) = x$$

- **ABS**: Element-wise absolute value operation.
$$\text{ABS}(x) = |x|$$

- **LOG**: Element-wise natural logarithm operation.
$$\text{LOG}(x) = \ln(x)$$

- **ABSLOG**: Element-wise absolute value of the natural logarithm operation.
$$\text{ABSLOG}(x) = |\ln(x)|$$

- **POW**: Element-wise power operation with a constant exponent.
$$\text{POW}(x, c) = x^c$$

- **EXP**: Element-wise exponential operation.
$$\text{EXP}(x) = e^x$$

- **NORMALIZE**: Normalization operation that scales the input to have a mean of 0 and a standard deviation of 1.
$$\text{NORMALIZE}(x) = \frac{x - \mu}{\sigma}$$
where $\mu$ is the mean of $x$, and $\sigma$ is the standard deviation of $x$.

- **SIGMOID**: Element-wise sigmoid function.
$$\text{SIGMOID}(x) = \frac{1}{1 + e^{-x}}$$

- **TANH**: Element-wise hyperbolic tangent function.
$$\text{TANH}(x) = \frac{e^x - e^{-x}}{e^x + e^{-x}}$$

## D.2 Reduction Operations $\mathcal{T}_{\text{reduce}}$

The reduction operations $\mathcal{T}_{\text{reduce}}$ aggregate the pre-processed scores into a single per-layer importance score $\mathcal{V}$. These operations use statistical measures and matrix operations to provide insights into the distribution and characteristics of the input data. The operations in this category include:

- **NO_OP**: No operation is performed, and the input is returned as is.

$$\text{NO\_OP}(x) = x$$

- **GRAM**: Gram matrix operation, which computes the matrix multiplication of the input with its transpose.

$$\text{GRAM}(X) = X^\top X$$

- **CORREF**: Correlation coefficient operation, which measures the linear relationship between the elements of the input.

$$\text{CORREF}(x) = \frac{x^\top x}{||x||_2^2}$$

- **DIAGONAL**: Diagonal operation, which extracts the diagonal elements of the input matrix.

$$\text{DIAGONAL}(X) = \text{diag}(X)$$

- **FROBENIUS_NORM**: Frobenius norm operation, which computes the square root of the sum of the squared elements of the input matrix.

$$\text{FROBENIUS\_NORM}(X) = \sqrt{\sum_{i,j} X_{i,j}^2}$$

- **L1_NORM**: L1 norm operation, which computes the sum of the absolute values of the elements in the input.

$$\text{L1\_NORM}(x) = \sum_i |x_i|$$

- **DETERMINANT**: Determinant operation, which computes the determinant of the input matrix.

$$\text{DETERMINANT}(X) = \det(X)$$

- **RANK**: Rank operation, which computes the rank of the input matrix.

$$\text{RANK}(X) = \text{rank}(X)$$

- **GEOMETRIC_MEAN**: Geometric mean operation, which computes the nth root of the product of the elements in the input.

$$\text{GEOMETRIC\_MEAN}(x) = \sqrt[n]{\prod_{i=1}^n x_i}$$

- **MEAN**: Mean operation, which computes the arithmetic mean of the elements in the input.

$$\text{MEAN}(x) = \frac{1}{n} \sum_{i=1}^n x_i$$

- **VAR**: Variance operation, which computes the variance of the elements in the input.

$$\text{VAR}(x) = \frac{1}{n} \sum_{i=1}^n (x_i - \mu)^2$$

where $\mu$ is the mean of $x$.

## D.3 Transform Operations $\mathcal{T}_{\text{trans}}$ and Post-process Operations $\mathcal{T}_{\text{post}}$

The transform operations $\mathcal{T}_{\text{trans}}$ map the per-layer importance scores $\mathcal{V}$ to a suitable range for sparsity ratios, while the post-process operations $\mathcal{T}_{\text{post}}$ ensure that the sparsity ratios $\phi$ satisfy any required constraints. The operations in these categories include:

- **NO_OP**: No operation is performed, and the input is returned as is.

$$\text{NO\_OP}(x) = x$$

- **SIN**: Sine function.

$$\text{SIN}(x) = \sin(x)$$

- **COS**: Cosine function.

$$\text{COS}(x) = \cos(x)$$

- **TAN**: Tangent function.

$$\text{TAN}(x) = \tan(x)$$

- **ASIN**: Inverse sine (arcsin) function.

$$\text{ASIN}(x) = \sin^{-1}(x)$$

- **ACOS**: Inverse cosine (arccos) function.

$$\text{ACOS}(x) = \cos^{-1}(x)$$

- **ATAN**: Inverse tangent (arctan) function.

$$\text{ATAN}(x) = \tan^{-1}(x)$$

- **EXP**: Exponential function.

$$\text{EXP}(x) = e^x$$

- **LOG**: Natural logarithm function.

$$\text{LOG}(x) = \ln(x)$$

- **ABS**: Absolute value function.

$$\text{ABS}(x) = |x|$$

- **SIGMOID**: Sigmoid function.

$$\text{SIGMOID}(x) = \frac{1}{1 + e^{-x}}$$

- **TANH**: Hyperbolic tangent function.

$$\text{TANH}(x) = \frac{e^x - e^{-x}}{e^x + e^{-x}}$$

