# OpenReview forum: "Discovering Sparsity Allocation for  Layer-wise Pruning of Large Language Models"
_NeurIPS.cc/2024/Conference — NeurIPS 2024 poster_

### Official Review · Reviewer_VeXf · 2024-07-10

**Soundness:** 2
**Presentation:** 2
**Contribution:** 2
**Rating:** 4
**Confidence:** 3

**Summary:**

This paper introduces DSA, an automated framework for determining layer-wise sparsity in large language models (LLMs). The approach aims to enhance pruning techniques by using an evolutionary algorithm to discover optimal sparsity allocation functions, thereby improving model performance on various tasks. The proposed method can improve the performance of layer-wise pruning baselines on various datasets.

**Strengths:**

**S1.**
This paper tackles an important problem in LLM efficiency by proposing a novel automated framework for sparsity allocation.

**S2.**
Extensive experiments demonstrate that DSA outperforms existing methods like SparseGPT and Wanda across multiple benchmarks and LLMs.

**S3.**
The approach is validated on diverse tasks, including arithmetic, knowledge reasoning, and multimodal tasks, showcasing its versatility and effectiveness.

**Weaknesses:**

**W1.**
The novelty of the approach is somewhat limited as it combines existing techniques such as AutoML and evolutionary algorithms, which are already well-explored in other contexts.

**W2.**
Some existing works, such as "ECoFLaP: Efficient Coarse-to-Fine Layer-Wise Pruning for Vision-Language Models" by Yi-Lin Sung et al., already address the same problem setting of adaptive sparsity allocation across layers, potentially diminishing the perceived novelty of this paper.

**W3.**
The methodology may require significant computational resources, which could be a limitation for practical applications.

**W4.**
The results on some benchmarks, while improved, may not be sufficiently groundbreaking to warrant acceptance in top-tier conferences like NeurIPS.

**Questions:**

**Q1.**
Can you provide more insights into the computational cost of the DSA framework compared to traditional pruning methods?

**Limitations:**

The authors discussed some limitations of their method in the conclusion.

---

> ### Author Rebuttal · Authors · 2024-08-06
>
> **Dear Reviewer VeXf ,**
>
> Thanks for the valuable feedback. We have tried our best to address all concerns in the last few days. If the reviewer finds our response adequate, **we would appreciate it if the reviewer considers raising the score**. Please see our responses below one by one：
>
>
>
> > **Q1: Compare to existing AutoML techniques.**
>
> **A1:** In contrast to common AutoML and evolutionary algorithms, our method is specifically **tailored for sparsity allocation discovery in LLMs** and involves several novel components:
>
> **(1) Novel problem formulation:** We are the first to frame LLM sparsity allocation as an AutoML problem, opening new avenues for optimizing LLM efficiency.
>
> **(2) Tailored search space:** We introduce a distinctive search space customized for LLM sparsity allocation, combining pre-processing, reduction, transformation, and post-processing operations in novel ways, allowing for more nuanced and effective sparsity distributions.
>
> **(3) Function discovery and generalization:** Diverging from typical AutoML methods such as NAS and HPO that search for specific modes or hyperparameters, our framework emphasizes generalized function discovery, identifying common patterns across LLMs and formuling  interpretable sparsity allocation.
>
> **(4) Search acceleration:** We develop LLM-specific acceleration techniques to reduce search time, making our DSA practical for large-scale LLM optimization.
>
> > **Q2: Compare to ECoFLaP [ref1] .**
>
> **A2:** We clarify that our novelty lies in being the **first automated search for adaptive sparsity methods**, which significantly differs from traditional adaptive pruning methods like ECoFLaP. Our work differs significantly from ECoFLaP in several key aspects：
>
> **(1) Automation and adaptability:** We employ an automated search method that eliminates the need for expert design and adapts strategies to different models and tasks, whereas ECoFLaP relies on hand-designed, hyperparameter tuning.
>
> **(2) Comprehensive search space:** Our comprehensive search space systematically maps   ***element-wise scores → per-layer importances → sparsity ratios***. In contrast, ECoFLaP simply  computes the keep ratio linearly during  its two-stage pruning.
>
> **(3)  Superior performance:** Our method obtains significant performance gains across various large language and multimodal models, demonstrating superior performance compared to ECoFLaP (see Table below).
>
> Table : Perplexity of Wanda, ECoFLaP and our DSA with LLaMA 7B at 0.6 sparsity  on WikiText2
>
> | Dense | Wanda | ECoFLaP(fIrst-order) | ECoFLaP(zeroth-order) | DSA (Ours) |
> | ----- | ----- | -------------------- | --------------------- | ---------- |
> | 7.26  | 10.68 | 10.16                | 9.83                  | **9.15**   |
>
> **We will augment this discussion and cite more Efficient AI studies [ref1]-[ref5] in the revision.**
>
>
>
> > **Q3 & Q5: About computational resources and cost compared to traditional methods**.
>
> **A3: Our responses are:**
>
> **(1)** Our method's main computational cost is in the initial search phase, taking about 0.5 days on LLaMA-1-7B. But the discovered allocation functions are transferable to other models **without additional costs. This one-time cost can be spread across multiple pruning runs.  As discussed in the limitations section.** we've developed acceleration techniques to address the search cost challenge common in AutoML methods.
>
> **(2)** Most conventional pruning methods also require time-consuming hyperparameter tuning, which may be less efficient than our well-optimized automated search process.
>
> **(3)** Applying our allocation function to prune the model proves highly efficient **(see following Table for detailed pruning speeds).** This efficiency stems from two key factors:  **(a)** we utilize element-wise scores from the uniform pruning method directly, avoiding extra forward and backward computations.  **(b)** we employ reduction operations to simplify computations linearly.
>
> *Table : Comparison of time overhead used for computing the pruning metric of LLaMA-1-65B to 50% sparsity.*
>
> | SparseGPT    | BESA      | ECoFLaP(Zeroth-order) | Wanda        | OWL(Wanda)  | DSA(Wanda)  |
> | ------------ | --------- | --------------------- | ------------ | ----------- | ----------- |
> | 1353 seconds | 4.5 hours | 6.6 seconds           | 5.6  seconds | 6.3 seconds | 6.5 seconds |
>
> > **Q4: About significance of results.**
>
> **A4: Our responses** are:
>
> **(1)** We would like to clarify that our method consistently achieves groundbreaking gains across multiple challenging tasks (reasoning and multimodal benchmarks). Notably, the LLaMA-1|2|3 model pruned by our DSA reach 7.48%|5.69%|14.14% gains on the state-of-the-art models Wanda and SparseGPT.
>
>  **(2)** To ensure fair comparisons, we follow standard settings like 50% sparsity, where baseline results are strong, making additional gains challenging.
>
>  **(3)** Our experiments in Table 8 at higher sparsity levels (65%-80%) on LLaMA-1 show our method's ability to achieve substantial gains.
>
>  **(4)** Results in the rebuttal PDF **(See our general response)** further demonstrate that our method can achieve substantial gains, with improvements ranging from **1% ~ 7% at 60% sparsity, 2% ~ 7.6% at 70% sparsity, and 1.4% ~ 4.2% at 2:4 sparsity.**
>
>
>
> > **References:**
> >
> > [ref1] ECoFLaP: Efficient Coarse-to-Fine Layer-Wise Pruning for Vision-Language Models. In ICLR, 2024.
> >
> > [ref2] Training Neural Networks with Fixed Sparse Masks. In NeurIPS, 2021.
> >
> > [ref3] Merge, Then Compress: Demystify Efficient SMoE with Hints from Its Routing Policy. In ICLR 2024.
> >
> > [ref4] LST: Ladder Side-Tuning for Parameter and Memory Efficient Transfer Learning. In NeurIPS 2022.
> >
> > [ref5] VL-Adapter: Parameter-Efficient Transfer Learning for Vision-and-Language Tasks. In CVPR 2022.
>
>
>
> **Finally,** we hope our response could address the concerns, and we thank the reviewer again for the helpful comments. We are glad to discuss further comments and suggestions.

---

> > ### Comment · Reviewer_VeXf · 2024-08-12
> >
> > Thank you for your thorough response to my initial review, and some of my previous issues have been resolved. I would like to follow up with a few additional questions:
> >
> > - You mentioned that the optimal allocation function requires approximately 0.5 days to compute, while this allocation function can be transferred to other models. Could you provide more empirical evidence supporting this claim? (apologize if I missed). Given that different LLMs can vary drastically in structure, depth, etc, how does it affect the generalizability of the allocation function? Furthermore, is it possible to transfer the allocation function between different pruning sparsity levels? If not, would it be necessary to redo the search for each sparsity setting?
> >
> > - The omission of a highly relevant paper, "*ECoFLaP: Efficient Coarse-to-Fine Layer-Wise Pruning for Vision-Language Models*" (ICLR, 2024), was a critical gap in the original submission. While I appreciate that you have now included an empirical comparison of your method with ECoFLaP, the current evaluation is limited to a single model, sparsity level, and dataset combination. Given the significance of this related work, I strongly encourage a more comprehensive and detailed comparison across various models, sparsity levels, and datasets to fully demonstrate the advantages of your approach.
> >
> > I apologize for the delay in my follow-up due to the heavy workload of the rebuttal process. I would greatly appreciate it if you could consider adding the additional experiments mentioned above.

---

> ### Author Response · Authors · 2024-08-11
> **Look Forward to The Post-Rebuttal Feedback**
>
> Dear Reviewer VeXf ,
>
> We express our deepest appreciation for your careful and constructive feedback. Our rebuttal addresses your concerns comprehensively, and we welcome any additional questions. If our response has successfully addressed your concerns and clarified the significance of our work, we would be immensely grateful if you could reconsider your recommendation. **We promise to add citations to related studies [ref1]-[ref5] in the revision.** We have diligently integrated feedback from all reviewers and hope this is positively reflected in your evaluation. Thank you for investing your valuable time in reviewing our response.
>
>
> With profound respect and gratitude,
>
> Paper 66 Authors

---

> ### Author Response · Authors · 2024-08-12
> **Request to review the rebuttal [Author-Reviewer discussion phase ending soon]**
>
> Dear Reviewer VeXf ,
>
> We would like to sincerely thank you again for your valuable feedback and insights, which have greatly improved our paper. We promise to thoroughly reflect all your comments in the final manuscript. As we are towards the end of the author-reviewer discussion period, we request you to please go through our rebuttal, and we would be immensely grateful if you could reconsider your recommendation.
>
> Best regards,
>
> Paper 66 Authors

---

> ### Author Response · Authors · 2024-08-13
> **Additional Response: Part 1**
>
> **Dear Reviewer VeXf ,**
>
> We would like to sincerely thank you again for your constructive feedback and apologize for any unclear points in the initial responses. After receiving new comments, we have made our best efforts to augment experiments and clarifications as follows. Please see our responses below one by one：
>
> ------
>
>
>
> > **Q6& Q7: Generalizability of allocation function and transferability between sparsity levels.**
>
> **A6:** We would like to highlight that all of our performance gains across models (LLaMA-1|2|3, Mistral, Vicuna, and OPT) and tasks (reasoning and multimodal benchmarks) are **obtained by transferring the same searched allocation function  on LLaMA-1-7B (see Equation 6), without additional individual searches:**
>
> **(1)  About empirical evidence:** **(a)** As **detailed in Table 2 &3 &4 & 5  [Line 250-286]**, our searched allocation function demonstrates high generality and consistent improvements across various model sizes and architectures, including LLaMA-1, LLaMA-2, LLaMA-3, OPT, Mistral, and LLaVA (Vicuna-based) models. For zero-shot accuracy at 50% sparsity in Table 2, our searched allocation function  shows significant gains, improving magnitude pruning by up to **14.14%, Wanda by up to 4.36%, and SparseGPT by up to 9.82% on LLaMA-3 8B**. On the MMLU task in Table 4, our searched allocation function shows  **0.96%  gains over OWL** for LLaMA-1 13B. In multimodal tasks in Table 6, our searched allocation function improves performance by up to **1.65% on LLaVA-1.5 7B and 1.98% on  LLaVA-1.5 13B (SQA task)  at 50% sparsity**. **(b)** We also directly transfer our searched allocation function on **LLaMA-3 70B** in the rebuttal PDF **(See our general response)**. Our  searched allocation function shows **2.23% to 4.38% gains** across different pruning approaches, aligning with our effectiveness on multiple LLMs.  **These results underscore the effectiveness and generality of our searched allocation function across various models, tasks.**
>
> **(2)  Understanding generalizability:** Sufficient experimental evidence in **(1)** solidly demonstrates the generality of our searched allocation function across different models and tasks. We provide several aspects to understand its generality: **(a)** In the model sparsity area, **different LLMs share the common sparsity laws** (e.g., important weights have salient gradients) resulting in sparsity methods with fixed metrics or functions that can be generalized to various models. For example, SparseGPT, Wanda, OWL and ECoFLaP also apply the same metric for different tasks without discussing whether their functions need to be changed for different structure and depth models (apologize if I missed). Our allocation function search space is **based on observation element-wise score distributions** in LLMs, as discussed in the **Introduction [Lines 62-78] and Figure 1(left).**  This observation aligns with OWL (see Figure 1 (middle)) and the sparsity laws that the initial layers of LLMs are more important. Because of these reasons, our searched allocation function naturally follows the sparsity laws, allowing good generalizability to the different LLMs.  **(b)** The operators contained in our allocation function are **architecture-agnostic and serve to normalize and eliminate architecture variance**. For example, our pre-process operator standardizes inputs by normalizing scores across layers, ensuring consistent performance metrics by addressing scale variations. As our **stability analyses  [Lines 243-250] and theoretical understanding in Appendix B [Lines 554-571]**, our allocation function is stable and can well alleviate perturbations of various models. In addition, as detailed in search robustness experiment (see Appendix C.1 [Lines 575-588]), our allocation functions searched with different initial seeds share the similar performance and similar expressions. **(c)**  **Based on (a) and (b), various LLMs share the common sparse allocation law and our search space and operations are architecture-agnostic, stable and robust to different search trials and models, resulting in good generality of our search allocation function.** To further validate this, we re-search the allocation function in OPT-6.7B model with different size and architecture. Our directly transferred allocation function has quite similar formulation and performance with the re-search function, confirming its generality across LLMs to some extent. We will involve this discussion in the revision.
>
> *Table: Mean accuracies (%)  of transferred and re-searched allocation functions with SparseGPT (Uniform baseline 55.19)  for OPT-6.7B at 0.5 sparsity*
>
> | Method      | Detailed Allocation Functions          | Mean accuracy (gain↑) |
> | ----------- | -------------------------------------- | --------------------- |
> | transferred | LOG _ABSLOG →GEOMETRIC_MEAN →COS →EXP  | 57.85 (2.66↑)         |
> | re-searched | LOG _ABSLOG →GEOMETRIC_MEAN →ACOS →SIGMOID | 57.96 (2.77↑)         |

---

> ### Author Response · Authors · 2024-08-13
> **Additional Response: Part 2**
>
> **(d)**  **In our A5 for Reviewer of38**, we apply our allocation functions to ConvNeXt in Table below. surpasses other methods, especially at higher sparsity levels, showing its generalizability for various models  **(Recognized by Reviewer of38 with improving the score)**.
>
> *Table: Accuracy (%) of Sparse ConvNeXt-Base on ImageNet-1K.*
>
> | Sparsity        | 50%       | 60%       | 70%       |
> | --------------- | --------- | --------- | --------- |
> | Wanda           | 82.72     | 80.55     | 68.18     |
> | OWL + Wanda     | 82.76     | 80.53     | 68.28     |
> | **DSA + Wanda** | **83.12** | **81.68** | **71.55** |
>
>
>
> **(3) About different sparsity levels:** **Yes.** We directly transfer searched  allocation function  (see Equation 6) for different sparsity levels and achieve noticeable gains at 65%-80% sparsity on LLaMA-1  in Table 8. In the rebuttal PDF **(See our general response)**, our searched  allocation function  demonstrates gains ranging from **1.24% to 7.03% at 60% sparsity and 1.91% to 7.68% at 70% sparsity on LLaMA-2 and on LLaMA-3**.
>
>
> > **Q8: About more comparison with ECoFLaP across various models, sparsity levels, and datasets.**
>
> **A8:** Following the suggestion, we conduct more experiments to compare our DSA method with ECoFLaP across various models, sparsity levels, and datasets. The results, summarized  in Table below, demonstrate the superior performance of our DSA method across different LLaMA models and sparsity rates. For the LLaMA-2-7B model, DSA achieves gains of **1.92% and 1.18% at 60% and 70% sparsity levels respectively, compared to ECoFLaP's gains of 0.77% and 0.52%**. Similar trends are observed for the LLaMA-2-13B model, where DSA outperforms ECoFLaP with improvements of 1.72% and **1.24% at 60% and 70% sparsity levels, surpassing ECoFLaP's gains of 0.86% and 0.65%**. The most significant improvements are seen in the LLaMA-3-70B model, where **DSA achieves remarkable gains of 2.23% and 2.34% at 60% and 70% sparsity levels, substantially outperforming ECoFLaP's gains of 0.71% and 0.68%**. These consistent improvements across different model sizes and sparsity levels highlight the effectiveness and generalizability of our DSA method, demonstrating its superiority over ECoFLaP in enhancing the performance of pruned language models.
>
> *Table: Mean accuracies (%) of our DSA on zero-shot task with 7 different datasets.*
>
> | Method              | LLaMA-2-7B (60%)  | LLaMA-2-7B (70%)  | LLaMA-2-13B (60%) | LLaMA-2-13B (70%) | LLaMA-3-70B (60%) | LLaMA-3-70B (70%) |
> | ------------------- | ----------------- | ----------------- | ----------------- | ----------------- | ----------------- | ----------------- |
> | Wanda               | 36.08             | 60.90             | 41.46             | 72.00             | 40.51             | 40.44             |
> | **Wanda (ECoFLaP)** | 36.85 (0.77↑)     | 61.42 (0.52↑)     | 42.32 (0.86↑)     | 72.65 (0.65↑)     | 41.22 (0.71↑)     | 41.12(0.68↑)      |
> | **Wanda (DSA)**     | **38.00 (1.92↑)** | **62.08 (1.18↑)** | **43.18 (1.72↑)** | **73.24 (1.24↑)** | **42.74 (2.23↑)** | **42.78 (2.34↑)** |
>
> ------
>
> **Finally,** we hope our response could address the concerns, and we thank the reviewer again for the helpful comments. **We promise to include these discussions and citations in the revision and sincerely hope that reviewer will consider improving the recommendation.**

---

> > ### Author Response · Authors · 2024-08-14
> > **Request to review the rebuttal [only a few hours left]**
> >
> > Dear Reviewer VeXf,
> >
> > We greatly appreciate the time and effort you've invested in reviewing our paper. Your constructive feedback has been invaluable in enhancing the quality of our work.
> >
> > We've diligently addressed all your concerns in our point-by-point rebuttal. As we approach the conclusion of the author-reviewer discussion period, we kindly request that you review our additional responses. Other two reviewers have raised their scores after considering our rebuttals and revisions. In light of this, we sincerely hope you'll also reconsider your recommendation and potentially improve your score.
> >
> > Once again, we extend our heartfelt thanks for your time and expertise. Your insights have been crucial to the refinement of our paper.
> >
> > Best regards,
> >
> > Paper 66 Authors

---

### Official Review · Reviewer_gYqu · 2024-07-11

**Soundness:** 3
**Presentation:** 3
**Contribution:** 3
**Rating:** 6
**Confidence:** 4

**Summary:**

This article introduces DSA (Discovering Sparsity Allocation) which is designed to automate the discovery of sparsity allocation schemes for layer-wise post-training pruning in large language models (LLMs).

**Strengths:**

1. This paper presents a framework for automatically discovering effective sparsity allocation functions.

2. This paper demonstrates consistent performance improvements across various tasks and datasets.

**Weaknesses:**

1. This article seems to combine OWL and Pruner-Zero. It uses the evolutionary algorithm to determine each layer's sparsity ratio. It's not quite innovative for me.

2. This article doesn't offer adequate evidence when the sparsity is higher than 50%. Table 12 shows the higher sparsity results on LLaMA1 7b. However, I think more evidence should be provided.


**Minor:**

L632:Typo of Erdos-Renyi.

The caption of Table 8: There is no 8B model in the LLaMA-1 family.

**Questions:**

1. This article only uses DSA to determine the sparsity ratio of each layer, while Pruner-zero searches for a pruning metric to prune the network. I am curious about the potential performance of DSA combined with Pruner-zero.

2. Can DSA be integrated with the 2:4 constraint to achieve actual acceleration?

3. The article mentions in line 227 that DSA can find potential allocation functions in only 0.5 days. What model was searched in this 0.5-day period, and how large was it?

**Limitations:**

The authors have addressed the limitations, but I believe they should also mention that current techniques do not support the inference of unstructured sparsity, which prevents them from achieving actual acceleration. While some companies are working on this, their devices are not yet publicly available.

---

> ### Author Rebuttal · Authors · 2024-08-06
>
> **Dear Reviewer gYq,**
>
> Thanks for the valuable feedback. We have tried our best to address all concerns in the last few days. If the reviewer finds our response adequate, **we would appreciate it if the reviewer considers raising the score**. Please see our responses below one by one:
>
> ------
>
> > **Q1: Compare to OWL and evolution method.**
>
> **A1:** Our method's novelty lies in being the **first automated search for adaptive sparsity allocation**, significantly differing from traditional adaptive pruning methods like OWL [14] and evolution methods like Pruner-Zero [37]:
>
> **(1) Compare to OWL:**
>
> **(a)** Our method revolutionizes the field with an automated search approach that removes the need for expert design.  In contrast, OWL relies on hand-designed, hyperparameter tuning and is limited by fixed form constraints.
>
> **(b)** Our comprehensive search space systematically maps element-wise scores to sparsity ratios, whereas OWL only computes the keep ratio based on outlier ratio linearly, lacking our introduced nonlinear mapping operators.
>
> **(c)** Comparative experiments show our method's significant outperformance over OWL in Table 8.
>
> **(2) Compare to evolution method:  Our DSA differs Pruner-Zero [37] in method type, search space, task, strategy, and Input-Output characteristics (See Table below).**
>
> **(a)** We uniquely frame LLM sparsity allocation as an AutoML challenge, opening novel avenues for enhancing LLM efficiency.
>
>  **(b)** Our search space is customized for LLM sparsity allocation, integrating various operations in innovative ways.
>
> **(c)** We develop LLM-specific acceleration techniques like program checking, making our approach practical for large-scale LLM optimization.
>
> | Method         | Types           | Task                             | Search space                                                | Input                          | Output                         | Strategy                   |
> | -------------- | --------------- | -------------------------------- | ----------------------------------------------------------- | ------------------------------ | ------------------------------ | -------------------------- |
> | Pruner-Zero    | uniform         | symbolic Pruning Metric          | unary/binary operations                                     | element-wise weights/gradients | element-wise score             | symbolic Regression        |
> | **DSA (ours)** | **non-uniform** | **adaptive allocation function** | **pre-process/reduction/transform/post-process operations** | **element-wise score**         | **layer-wise sparsity ratios** | **evolutionary Algorithm** |
>
> Please note that the title and abstract of Pruner-Zero [37] are accessible in May 2024, **but its full paper is released (on arXiv and OpenReview) in June 2024, after the NeurIPS deadline.** We already discuss OWL and Pruner-Zero in the related work [lines 143-148] and will add more comparisons in the revision.
>
> > **Q2: More results for high sparsity ratios.**
>
> **A2: Our responses** are:
>
> **(1)** To ensure fair comparisons, we follow standard settings like 50% sparsity, where baseline results are strong, making additional gains challenging.
>
> **(2)** Our experiments in Table 8 at higher **sparsity levels (65%-80%)** on LLaMA-1 show our method's ability to achieve substantial improvements.
>
>  **(3)** Following the suggestions, we provide 60% and 70% sparse experiments in the rebuttal PDF **(See our general response)**. Our method demonstrates **gains ranging from 1.24% to 7.03% in LLaMA-2-7B&13B, and LLaMA-3-70B at 60% sparsity. At 70% sparsity, our improvements ranging from 1.91% to 7.68%** underscore the effectiveness of our DSA for high sparsity ratios.
>
> > **Q3: About typos.**
>
> **A3:** We appreciate the reviewer's corrections of typos in Erdos-Renyi and LLaMA-1-7B and commit to fixing them in the revision.
>
> > **Q4: About potential combination with Pruner-Zero.**
>
> **A4: Yes.** Following the suggestion, we conduct experiments combining our method with Pruner-Zero, yielding new state-of-the-art results. This successful integration is due to the orthogonal nature of the two methods: Pruner-Zero optimizes element-wise importance scoring, while our DSA specializes in adaptive layer-wise sparsity allocation.
>
> *Table: Mean accuracies (%) at 0.5 sparsity on 7 zero-shot task.*
>
> | Models      | Pruner-zero | **Pruner-zero+ DSA (Ours)** |
> | ----------- | ----------- | ----------------------- |
> | LLaMA-2-7B  | 58.87       | **62.22**                   |
> | LLaMA-2-13B | 64.83       | **67.05**                  |
>
> > **Q4: About integration with 2:4 constraint.**
>
> **A4:**  Following the suggestions, we evaluate our DSA under 2:4 sparsity on **LLaVA (See Table below) and LLaMA-2  (See our general response)**. These consistent gains **(2.2%  ~ 2.8% on LLaVA and 1.4% ~ 4.2% on LLaMA-2)**  across different model sizes and datasets suggest that our approach is more effective at maintaining performance under tighter sparsity constraints.
>
> | 7B LLaVA-1.5   | VQAv2     | SQA       | VQA       | 13B LLaVA-1.5  | VQAv2     | SQA       | VQA       |
> | -------------- | --------- | --------- | --------- | -------------- | --------- | --------- | --------- |
> | Dense          | 78.50     | 66.80     | 58.20     | Dense          | 80.00     | 74.94     | 61.30     |
> | Wanda (2:4)    | 68.92     | 55.06     | 45.42     | Wanda (2:4)    | 75.39     | 64.89     | 52.52     |
> | **Ours (2:4)** | **71.18** | **57.44** | **48.25** | **Ours (2:4)** | **76.75** | **67.13** | **54.05** |
>
>
>
> ------
>
> > **Q5: Details of  the time cost .**
>
> **A5:** As **detailed in the experiment [Line 252-258]**, the 0.5-day search time mentioned is for the **LLaMA-1 7B model** on WikiText2 based on Wanda at 50% sparse setting. We will clarify this detail in the revision.
>
>
> ------
>
>
> **Finally,** we hope our response could address the concerns, and we thank the reviewer again for the helpful comments. We are glad to discuss further comments and suggestions.

---

> ### Author Response · Authors · 2024-08-11
> **Look Forward to The Post-Rebuttal Feedback**
>
> Dear Reviewer gYqu,
>
> We are profoundly grateful for your thorough and constructive comments. We have addressed your concerns point-by-point in our rebuttal and welcome any further inquiries. If our response has successfully alleviated your concerns and highlighted the merit of our work, we would be deeply appreciative if you could reconsider your recommendation.  We promise to thoroughly reflect all your comments in the final manuscript. We have conscientiously incorporated feedback from all reviewers and hope this is positively reflected in your assessment. Thank you for generously dedicating your time to review our response.
>
> With sincere gratitude,
>
>
>
> Paper 66 Authors

---

> ### Author Response · Authors · 2024-08-12
> **Request to review the rebuttal [Author-Reviewer discussion phase ending soon]**
>
> Dear Reviewer gYqu,
>
> Thank you again for spending your valuable time reviewing our paper. We have carefully considered and addressed your concerns. As we are towards the end of the author-reviewer discussion period, we request you to please go through our rebuttal. We appreciate your constructive and insightful comments and will definitely incorporate them into the final manuscript.
>
> Best regards,
>
> Paper 66 Authors

---

> > ### Author Response · Authors · 2024-08-13
> > **Request to review the rebuttal [only one day left]**
> >
> > Dear Reviewer gYqu,
> >
> >
> > Thank you for your time reviewing the paper. Your constructive feedback will help improve the quality of our paper.
> >
> > We have also addressed all your concerns in our rebuttal point-by-point. As we are towards the end of the author-reviewer discussion period, we request you to please go through our rebuttal, and we sincerely hope that the reviewer will consider improving the recommendation.
> >
> > We thank you again for your time!
> >
> > Best regards,
> >
> > Paper 66 Authors

---

> ### Comment · Reviewer_gYqu · 2024-08-13
> **Official Comment by Reviewer gYqu**
>
> Thanks for the authors' replies. Most of my questions have been addressed.
>
> One remaining question is about the running time. As the author replied to Reviewer s93u: For other different LLM models, we directly transfer this discovered allocation function without additional search.
>
> I'm wondering how this is realized in practice. Given a different model, for instance, the LLaMA2-70b or OPT model family, how can it be transferred into the new model which has different number of layers or model structure?

---

> ### Author Response · Authors · 2024-08-13
> **Additional Response**
>
> **Dear Reviewer gYq,**
>
> We would like to express our sincere gratitude for your time and valuable feedback on our paper.
> In our rebuttal, we have carefully considered and incorporated your suggestions. We are eagerly anticipating any additional feedback you may have.
> Additionally, to further address your concerns regarding **generalizability of allocation function in the running time**, we make best efforts to clarify comprehensively as follows:
>
> **(1) Practical implementations** of searched allocation function: The searched allocation function comprises four operations: LOG_ABSLOG (pre-process), GEOMETRIC_MEAN (reduction), COS (transformation), and EXP (post-process).   **(a)** Among them, our pre-process operation is element-wise and architecture-agnostic, similar to metrics in SparseGPT or Wanda.  Our reduction operation, akin to OWL's outlier operator, processes intra-layer scores to importances, independent from layer lengths and generalize to various  model structures.  **(b)**  As alternative to OWL linear expression, our transformation post-process operations are non-parametric functions and  convert  per-layer importances → sparsity ratios. **They (implemented as "COS" and "EXP" using PyTorch's torch.cos and torch.exp functions) can handle inputs of different lengths (corresponding to different layer counts), making them adaptable to diverse model sizes.**  **(c)**  In summary, our searched allocation function can be applied like OWL to different numbers of layers or model structures. We have also provided the implementation codes in the Supplementary Material and will open-source them upon acceptance. The core practical code of our searched allocation function is in the Supplementary Material's code\lib\autolayer.py file.
>
> **(2)  Understanding generalizability:** Sufficient **results in Table 2 &3 &4 & 5  [Line 250-286] and rebuttal PDF (See our general response)** solidly demonstrates the generality of our allocation functions across different models and tasks. We provide several aspects to understand its generality: **(a) Different LLMs share the common sparsity laws** (e.g., important weights have salient gradients) resulting in sparsity methods with fixed metrics that can be generalized to various models. For example, Wanda and OWL apply the same metric for different tasks without discussing whether their functions need to be changed for different structure and depth models. Our allocation function search space is **based on observation element-wise score distributions** in LLMs, as discussed in the **Introduction [Lines 62-78] and Figure 1(left).**  This observation aligns with OWL and the sparsity laws that the initial layers of LLMs are more important. Thus, our searched allocation function naturally follows the sparsity laws, allowing good generalizability to the different LLMs.  **(b)** The operators contained in our allocation function are **architecture-agnostic and serve to normalize and eliminate architecture variance**. For example,  our pre-process operator standardizes inputs by normalizing scores across layers, ensuring consistent performance metrics by addressing scale variations. As our **stability analyses  [Lines 243-250] and theoretical understanding in Appendix B [Lines 554-571]**, our allocation function is stable and can well alleviate perturbations of various models. As detailed in search robustness experiment (see Appendix C.1 [Lines 575-588]), our allocation functions searched with different initial seeds share  similar performance and  expressions. **(c)**  **Based on (a) and (b), various LLMs share the common sparse allocation law and our search space and operations are architecture-agnostic, stable and robust to different search trials and models, resulting in good generality of our search allocation function.** To further validate this, we re-search the allocation function in OPT-6.7B model with different size and architecture. Our directly transferred allocation function has quite similar formulation and performance with the re-search function, confirming its generality across LLMs to some extent. We will involve this discussion in the revision.
>
> *Table: Mean accuracies (%)  of transferred and  re-searched allocation functions with SparseGPT (Uniform baseline 55.19)  for OPT-6.7B at 0.5 sparsity*
>
> | Method      | Detailed Allocation Functions          | Mean accuracy (gain↑) |
> | ----------- | -------------------------------------- | --------------------- |
> | transferred | LOG _ABSLOG →GEOMETRIC_MEAN →COS →EXP  | 57.85 (2.66↑)         |
> | re-searched | LOG _ABSLOG →GEOMETRIC_MEAN →Sine →EXP | 57.96 (2.77↑)         |
>
> -----
>
>  **Finally,** we hope our response could address the concerns, and we thank the reviewer again for the helpful comments. **We promise to include these discussions in the revision and sincerely hope that reviewer will consider improving the recommendation.**
>
>
>
> Best regards,
>
>
>
> Paper 66 Authors

---

> > ### Comment · Reviewer_gYqu · 2024-08-13
> > **Official Comment by Reviewer gYqu**
> >
> > Thanks for your replies. Most of my concerns are addressed. I will raise my score.

---

> > > ### Author Response · Authors · 2024-08-14
> > > **Many Thanks for the Increasingly Positive Assessment and Recognition of Our Work and Rebuttal**
> > >
> > > Dear Reviewer gYq,
> > >
> > > We are profoundly thankful for your thoughtful reconsideration of our work and rebuttal. Your decision to raise the rating is not only deeply appreciated but also serves as a powerful source of motivation for us.
> > >
> > > Your recognition of our efforts is incredibly heartening and reinforces our dedication to excellence. We are committed to leveraging your valuable insights to further refine and elevate the quality of our paper.
> > >
> > > We cannot express enough gratitude for your constructive comments, the time you've invested, and your patience throughout this review process. Your expertise and guidance have been absolutely crucial in enhancing the depth and rigor of our research.
> > >
> > > We are truly indebted to you for your thorough evaluation and genuine engagement with our responses. Your significant contribution to the peer review process exemplifies the highest standards of academic integrity and collaboration.
> > >
> > > With heartfelt appreciation and warmest regards,
> > >
> > > Paper 66 Authors

---

### Official Review · Reviewer_s93u · 2024-07-12

**Soundness:** 3
**Presentation:** 3
**Contribution:** 2
**Rating:** 6
**Confidence:** 2

**Summary:**

This paper presents DSA, which models layer importance to sparsity ratios, and integrates the allocation function discovered by evolutionary algorithms into various methods, resulting in significant performance improvements.

**Strengths:**

This manuscript is a qualified paper, i.e,
The method seems technically sound and straightforward in principle.
Empirical results demonstrate the strength of this approach.

**Weaknesses:**

All the experiments in this article are done on unstructured sparsity. It is well known that unstructured sparsity will not bring the effect of computation acceleration. Have the authors ever tried the experimental results on structured sparsity?

It is well known that search algorithms are often very time-consuming, so the 0.5day mentioned by L227 is the result of experiments conducted on which model, 7B, 13B and 70B?

**Questions:**

Please see weaknesses

---

> ### Author Rebuttal · Authors · 2024-08-06
>
> **Dear Reviewer s93u,**
>
> Thanks for constructive comments.  We have tried our best to address all concerns in the last few days. If the reviewer finds our response adequate, **we would appreciate it if the reviewer considers raising the score**. Please see our responses below one by one：
>
>
>
> ------
>
> > **Q1: About results on structured sparsity.**
>
> **A1: Our responses** are: **(1)** Following the suggestion, we apply our non-uniform layer-wise sparsity allocation method to the structured pruning technique LLM Pruner [32], which accelerates pruned LLMs directly. The results in Table below show that **our  method performs well in structured pruning scenarios** and outperforms OWL.
>
> *Table: Perplexity of Structure Pruning with LLaMA-7B on WikiText-2.*
>
> | Pruning Method | Layerwise Sparsity | 20%       | 40%       | 60%       | 80%        |
> | -------------- | ------------------ | --------- | --------- | --------- | ---------- |
> | LLM Pruner     | Uniform            | 19.09     | 30.39     | 90.017    | 1228.17    |
> | LLM Pruner     | OWL                | 18.57     | 28.65     | 76.99     | 321.64     |
> | **LLM Pruner** | **DSA**            | **17.85** | **26.98** | **68.82** | **202.42** |
>
>
>
> **(2)** We would like to highlight that our method **supports and enhances 2:4 and 4:8 structured sparsity**, compatible with Nvidia GPUs for hardware acceleration. The results presented in Tables 6 and 7, as well as the results in the rebuttal PDF **(See our general response)**, demonstrate the efficacy of our approach, showcasing **gains of 1.4% ~ 4.2% at 2:4 sparsity and ~1% gains at 4:8 sparsity.** These gains are promising advancements in this area **(Recognized by Reviewer 1f5c, of38, gYqu, and VeXf)**.
>
> **(3)** Recent advancements in advanced GPU kernels like NVIDIA cuSPARSE [ref1], Sputnik [ref2], and Flash-LLM [ref3] have rapidly supported unstructured sparsity. This practical relevance extends beyond GPUs to non-GPU hardware such as CPUs (e.g., XNNPACK [ref4]) and specialized accelerators like FPGA accelerator. Our method under unstructured sparsity also **achieves 1.8x~3.7x speed up  with  DeepSparse inference engine** for LLaMA-V2-7B-chat-hf, as presented in Table below.
>
> | Method   | Unstructured Sparsity   | Dense    | 40%      | 50%      | 60%      | 70%      | 80%      |
> | -------- | ----------------------- | -------- | -------- | -------- | -------- | -------- | -------- |
> | ours     | Latency (ms)            | 213.8    | 121.4    | 90.8     | 86.0     | 78.3     | 58.6     |
> | ours     | Throughput (tokens/sec) | 4.7      | 8.2      | 11.0     | 11.6     | 12.8     | 17.1     |
> | **ours** | **Speedup**             | **1.0x** | **1.8x** | **2.4x** | **2.5x** | **2.7x** | **3.7x** |
>
>
>
> ------
>
> > **Q2: Details of  the time cost .**
>
> **A2:** As **detailed in the experiment [Line 252-258],**  the 0.5-day search time mentioned is for the **LLaMA-1 7B model** on WikiText2 based on Wanda at 50% sparse setting.  **For other different LLM models, we directly transfer this discovered allocation function without additional search.** We apologize for not explicitly stating this and will clarify this detail in the revision.
>
>
>
> > **References:**
> >
> > [ref1] NVIDIA GPUs scalability to solve multiple (batch) tridiagonal systems implementation of cuThomasBatch. In PPAM 2018.
> >
> > [ref2] Sparse gpu kernels for deep learning. In SC conference 2020.
> >
> > [ref3] Flash-LLM: Enabling Cost-Effective and Highly-Efficient Large Generative Model Inference with Unstructured Sparsity. ArXiv:2309.10285.
> >
> > [ref4] Fast sparse convnets. In CVPR2020.
>
>
>
> **Finally,** we hope our response could address the concerns, and we thank the reviewer again for the helpful comments. We are glad to discuss further comments and suggestions.

---

> ### Author Response · Authors · 2024-08-11
> **Look Forward to The Post-Rebuttal Feedback**
>
> Dear Reviewer s93u,
>
> We are deeply thankful for your careful consideration and constructive feedback. Our rebuttal addresses your concerns in detail, and we are open to any additional questions. If our response has successfully clarified the value of our work and addressed your concerns, we would be incredibly grateful if you could reconsider your recommendation. We promise to thoroughly reflect all your comments in the final manuscript. We have diligently incorporated feedback from all reviewers and hope this effort is evident in your evaluation. Thank you for your time and expertise in reviewing our response.
>
> With utmost respect,
>
> Paper 66 Authors

---

> ### Author Response · Authors · 2024-08-12
> **Request to review the rebuttal [Author-Reviewer discussion phase ending soon]**
>
> Dear Reviewer s93u,
>
> Thank you for your time reviewing the paper. Your constructive feedback will help improve the quality of our paper.
>
> We have also addressed all your concerns in our rebuttal point-by-point. As we are towards the end of the author-reviewer discussion period, we request you to please go through our rebuttal, and we would be truly grateful if you could reconsider your recommendation.
>
> We thank you again for your time!
>
> Best regards,
>
> Paper 66 Authors

---

> > ### Author Response · Authors · 2024-08-13
> > **Request to review the rebuttal [only one day left]**
> >
> > Dear Reviewer s93u,
> >
> >
> > Thank you for your time reviewing the paper. Your constructive feedback will help improve the quality of our paper.
> >
> > We have also addressed all your concerns in our rebuttal point-by-point. As we are towards the end of the author-reviewer discussion period, we request you to please go through our rebuttal, and we sincerely hope that the reviewer will consider improving the recommendation.
> >
> > We thank you again for your time!
> >
> > Best regards,
> >
> > Paper 66 Authors

---

> ### Author Response · Authors · 2024-08-14
> **Request to review the rebuttal [only a few hours left]**
>
> Dear Reviewer s93u,
>
> We greatly appreciate the time and effort you've invested in reviewing our paper. Your constructive feedback has been invaluable in enhancing the quality of our work.
>
> We've diligently addressed all your concerns in our point-by-point rebuttal. As we approach the conclusion of the author-reviewer discussion period, we kindly request that you review our responses. Other two reviewers have raised their scores after considering our rebuttals and revisions. In light of this, we sincerely hope you'll also reconsider your recommendation and potentially improve your score.
>
> Once again, we extend our heartfelt thanks for your time and expertise. Your insights have been crucial to the refinement of our paper.
>
> Best regards,
>
> Paper 66 Authors

---

### Official Review · Reviewer_of38 · 2024-07-14

**Soundness:** 3
**Presentation:** 3
**Contribution:** 3
**Rating:** 6
**Confidence:** 4

**Summary:**

This paper introduces DSA, an automated framework for discovering optimal sparsity allocation schemes for layer-wise pruning in LLMs. The proposed framework uses per-layer importance statistics and an evolutionary algorithm to explore effective allocation functions, which are then integrated into various pruning methods. Extensive experiments on challenging tasks demonstrate significant performance gains for models like LLaMA-1|2|3, Mistral, and OPT, achieving notable improvements over state-of-the-art models.

**Strengths:**

1. The authors evaluated on a wide range of models, ranging from representative LLMs like LLaMA-1/2/3, Mistral, OPT to a multi-modal model LLaVA.

2. This paper tackles an important problem as to how to assign layerwise pruning ratio for sparsity based pruning. The authors have demonstrated strong results compared to state-of-the-art methods.

**Weaknesses:**

1. A majority of the results in this paper are performed under the setting of 50% unstructured sparsity, which is a relatively low sparsity level. It would be good to demonstrate more results on higher level of sparsity, e.g., 60% and 70%.

2. The authors described the search space of the proposed algorithm as four transform operations in Equation 3. However, I am not sure if this is the best way to find layerwise sparsity. Is there any motivation for such design of the search space? More specifically, why should we convert the element-wise scores to layer-wise scores first? I think some space should be used to discuss the motivation and insights behind Equation 3.

3. It would be good to show some numbers on the practical runtime speedup of dynamic layerwise pruning as compared to layerwise uniform pruning.

**Questions:**

1. Are the top-performing allocation functions the same across pruning settings, e.g., sparsity ratio and models? More specifically, does the allocation function in equation 6 generalize across LLMs?

2. How does the proposed sparsity allocation method apply to neural networks beyond Transformers, e.g., convolutional neural networks?

3. The authors evaluated all the open-source LLMs in LLaMA and LLaMA-2. However, for LLaMA-3, only the 8b model is evaluated. Have the authors experimented with LLaMA-3-70B?

**Limitations:**

Yes.

---

> ### Author Rebuttal · Authors · 2024-08-06
>
> **Dear Reviewer of38,**
>
> Thanks for constructive comments.  We have tried our best to address all concerns in the last few days. If the reviewer finds our response adequate, **we would appreciate it if the reviewer considers raising the score**. Please see our responses below one by one：
>
>
>
> ------
>
> > **Q1: More results on higher sparsity levels.**
>
> **A1: Our responses** are:
>
> **(1)** To ensure fair comparisons, we follow standard settings like 50% sparsity, where baseline results are strong, making additional gains challenging.
>
>  **(2)** Our experiments in **Table 8 at higher sparsity levels (65%-80%) on LLaMA-1** show our method's ability to achieve substantial improvements **(Recognized by Reviewer s93u, gYqu, and VeXf)**.
>
> **(3)** Following the suggestions, we provide 60% and 70% sparse experiments in the rebuttal PDF **(See our general response).**  Our method demonstrates gains ranging from **1.24% to 7.03% at 60% sparsity and 1.91% to 7.68% at 70% sparsity**, which underscore our effectiveness for high sparsity ratios.
>
> -----
>
> > **Q2: About motivation and insights of design of the search space**.
>
> **A2: Our responses** are:
>
> **(1)** As discussed in **Introduction [Lines 62-78] and Figure 1 (left),** our design is motivated by analyzing element-wise score distributions, and mean values of per-layer scores, inspiring reduction and transformation ops. While basic reduction showed modest gains, applying transform ops yielded more promising results. We also include pre-processing for score normalization and post-processing to enhance function fit.
>
>  **(2) Converting element-wise scores to layer-wise scores is important:** layer-wise scores provide a consolidated view of importance, reducing noise and offering more stable ratios. By focusing on layer-wise metrics, critical layer information can be leveraged for better parameter retention. This process enhances computational efficiency by simplifying complexity from the element to the layer level.
>
>  **(3)**  **As detailed in Primary Operators section [Lines 184-197]**, key insights of our design are:
>
> **(a) Pre-process:** Standardizes inputs by normalizing scores across layers, ensuring consistent performance metrics by addressing scale variations.
>
>  **(b) Reduction:** Condenses element-wise information by extracting representative values per layer, reducing computational complexity.
>
>  **(c) Transformation**: Models complex relationships with functions, enabling the representation of intricate patterns in layer importance.
>
> **(d) Post-process:** Fine-tunes the allocation function for optimization, enhancing flexibility.
>
>  **(4)**  Ablation study  **(See Table below)**  shows that the Reduction ops is the most influential, followed by the Transformation ops.
>
> *Table : Perplexity with LLaMA 7B at 0.7 sparsity on WikiText2*
>
> | DSA   | DSA without Pre-process | DSA without Reduction | DSA without Transformation | DSA without Post-process |
> | ----- | ----------------------- | --------------------- | -------------------------- | ------------------------ |
> | 22.60 | 23.45                   | 26.55                 | 25.61                      | 23.22                    |
>
>
>
> > **Q3: About practical runtime speedup.**
>
> **A3:** Following the suggestions, we evaluate the runtime speedup   **(See Table below)**  of employing DeepSparse inference engine on LLaMA-V2-7B-chat-hf model. The findings indicate that our method exhibits comparable speedup to uniform pruning.
>
> | Method   | Sparsity               | Dense    | 40%      | 50%      | 60%      | 70%      | 80%      |
> | -------- | ---------------------- | -------- | -------- | -------- | -------- | -------- | -------- |
> | uniform  | Latency (ms)            | 213.8    | 112.5    | 89.1     | 85.5     | 82.2     | 61.1     |
> | uniform  | Throughput (tokens/sec) | 4.7      | 8.9      | 11.2     | 11.7     | 12.2     | 16.4     |
> | uniform  | Speedup                | 1.0x     | 1.9x     | 2.4x     | 2.5x     | 2.6x     | 3.5x     |
> | ours     | Latency (ms)            | 213.8    | 121.4    | 90.8     | 86.0     | 78.3     | 58.6     |
> | ours     | Throughput (tokens/sec) | 4.7      | 8.2      | 11.0     | 11.6     | 12.8     | 17.1     |
> | **ours** | **Speedup**            | **1.0x** | **1.8x** | **2.4x** | **2.5x** | **2.7x** | **3.7x** |
>
> > **Q4: About generalization of allocation functions.**
>
> **A4: Yes.**  We directly transfer top-performing allocation function in Equation 6 to different models and takes without additional search. Our experiments show this allocation functions do show good generalization across models and sparsity settings.
>
> > **Q5: About applicability to non-Transformer networks.**
>
> **A5:** We would like to highlight that **our method is architecture-agnostic and can be applied to various model types, including CNNs. The reason is that our search space is comprehensive and includes diverse types of operations that can be utilized across different model architectures.** To confirm this, we apply our allocation functions to ConvNeXt in Table below.  Our DSA surpasses other methods, especially at higher sparsity levels, showing its effectiveness with CNNs.
>
> *Table: Accuracy (%) of Sparse ConvNeXt-Base on ImageNet-1K.*
>
> | Sparsity        | 50%       | 60%       | 70%       |
> | --------------- | --------- | --------- | --------- |
> | Wanda           | 82.72     | 80.55     | 68.18     |
> | OWL + Wanda     | 82.76     | 80.53     | 68.28     |
> | **DSA + Wanda** | **83.12** | **81.68** | **71.55** |
>
> > **Q6: About experiment with on LLaMA-3 70B.**
>
> **A6:** Following the suggestion, we perform experiments on LLaMA-3 70B in the rebuttal PDF **(See our general response)**. Our method shows **2.23% to 4.38% gains** across different pruning approaches, aligning with our effectiveness on multiple LLMs.
>
>
>
> **Finally,** we hope our response could address the concerns, and we thank the reviewer again for the helpful comments. We are glad to discuss further comments and suggestions.

---

> ### Author Response · Authors · 2024-08-11
> **Look Forward to The Post-Rebuttal Feedback**
>
> Dear Reviewer of38,
>
> We extend our heartfelt thanks for your meticulous and invaluable comments. Our rebuttal addresses your concerns comprehensively, and we welcome any further inquiries you may have. If our response alleviates your concerns and clarifies the value of our paper, we would be truly grateful if you could reconsider your recommendation. We pledge to thoroughly incorporate all your astute observations in the final version. We have conscientiously integrated feedback from all reviewers and hope this is reflected favorably in your assessment. We are truly grateful for the time you've invested in reviewing our work.
>
> With sincere appreciation,
>
>
> Paper 66 Authors

---

> ### Author Response · Authors · 2024-08-12
> **Request to review the rebuttal [Author-Reviewer discussion phase ending soon]**
>
> Dear Reviewer of38,
>
> Thank you for your time reviewing the paper. Your constructive feedback will help improve the quality of our paper.
>
> We have also addressed all your concerns in our rebuttal point-by-point. As we are towards the end of the author-reviewer discussion period, we request you to please go through our rebuttal, and we would be truly grateful if you could reconsider your recommendation.
>
> We thank you again for your time!
>
> Best regards,
>
> Paper 66 Authors

---

> > ### Comment · Reviewer_of38 · 2024-08-12
> >
> > I would like to thank the authors for the response. My concerns are adequately addressed. Thus I have improved my score to 6.

---

> > > ### Author Response · Authors · 2024-08-12
> > > **Many thanks for improving the score and recognition of our work and rebuttal**
> > >
> > > Dear Reviewer of38,
> > >
> > > Thank you so much for the recognition of our responses. We are glad to see that you have raised your score.
> > >
> > > We will make more efforts to improve our paper further.
> > >
> > > Many thanks for your constructive comments, time and patience.
> > >
> > >
> > >
> > > Best regards and thanks,
> > >
> > >
> > > Paper 66 Authors

---

### Official Review · Reviewer_1f5c · 2024-07-15

**Soundness:** 3
**Presentation:** 2
**Contribution:** 3
**Rating:** 5
**Confidence:** 2

**Summary:**

This paper introduce a novel model pruning algorithm, DSA, aiming to prune unimportant model weights to increase the sparsity of models. Unlike previous model pruning methods which assign the same sparsity ratio for all layers. DSA proposes a method to calculate the sparsity allocation for different layers, which achieves a more adaptive pruning for each layer. The evaluation shows that DSA has strong empirical performance.

**Strengths:**

1. The insight of the paper is clear and easy to understand.
2. The paper proposes a novel method to estimate the allocation budget for different layers.
3. The proposed method shows better performance than SOTA pruning methods.

**Weaknesses:**

1. The presentation of the proposed method is quite confusing. Especially the section 4, I am not quite clear about why the allocation function uses this design. Why do we need the pre-process, reduction, transformation, and post-process? What are the insights for these components?
2. Performance improvement is not as promising as the abstract claims, especially in structured pruning cases. Structured pruning is a more meaningful setting for model pruning, since it can be directly accelerated by hardware to achieve wall-clock time speedup. In Tables 6 and 7, the performance improvement is around 1%.
3. Wanda paper also reported 2:4 sparsity. How is the performance of DSA in a 2:4 sparsity setting?

**Questions:**

See weakness.

**Limitations:**

Yes

---

> ### Author Rebuttal · Authors · 2024-08-06
>
> **Dear Reviewer 1f5c,**
>
> Thanks for the valuable feedback. We have tried our best to address all concerns in the last few days. If the reviewer finds our response adequate, **we would appreciate it if the reviewer considers raising the score**. Please see our responses below one by one：
>
> ------
>
> > **Q1: About motivation and insights of allocation function design.**
>
> **A1: Our responses** are:
>
> **(1)** Our allocation function design is **motivated by analyzing element-wise score distributions**, as discussed in the **Introduction [Lines 62-78] and Figure 1(left).**
>
> **(a)** We notice that mean, variance, and entropy values of per-layer element-wise scores can serve as allocation indicators, inspiring reduction operations.
>
> **(b)** While basic reduction of element-wise scores showed modest improvements, applying transform functions yielded more promising results, prompting the introduction of transform operations.
>
> **(c)** We include pre-process to normalize scores for fair comparison and post-process to further enhance function fit's upper bound.
>
>  **(2)**  The key insight of our four-component design is its flexibility in exploring diverse allocation functions tailored to each LLM's characteristics, while capturing complex, non-linear relationships between element-wise scores and sparsity ratios. **As detailed in the Primary Operators section [Lines 184-197]**, each component serves a specific purpose:
>
>  **(a) Pre-process:** Standardizes inputs by normalizing scores across layers, ensuring consistent performance metrics by addressing scale variations.
>
>  **(b) Reduction:** Condenses element-wise information by extracting representative values per layer through operations like variance or entropy, reducing computational complexity.
>
>  **(c) Transformation**: Models complex relationships with functions like sine or exponential, enabling the representation of intricate patterns in layer importance.
>
>  **(d) Post-process:** Fine-tunes the allocation function for optimization, enhancing flexibility.
>
>  **(3)** Ablation study  **(See Table below)**  shows that the Reduction component is the most influential, followed by the Transformation component, while the Pre-processing and Post-processing components have a smaller impact on the overall DSA performance.
>
> *Table : Perplexity  with LLaMA 7B at 0.7sparsity  on WikiText2*
>
> | DSA   | DSA without Pre-process | DSA without Reduction | DSA without Transformation | DSA without Post-process |
> | ----- | ----------------------- | --------------------- | -------------------------- | ------------------------ |
> | 22.60 | 23.45                   | 26.55                 | 25.61                      | 23.22                    |
>
> ------
>
> > **Q2: About performance improvements.**
>
> **A1: Our responses** are:
>
>  **(1)** We would like to clarify that our method consistently demonstrates substantial enhancements **(1%-14%) across a spectrum of scales (7B~70B), models (LLaMA-1|2|3, Mistral, LLaVA, and OPT models), and complex tasks (reasoning and multimodal benchmarks)**,  which robustly confirms its effectiveness.  The magnitude of gains naturally varies depending on the model and task. For instance, in Tables 6 & 7, considering the **Dense LLaVA-1.5 Models ranging from 7B to 13B, we observe a only 1.5%** performance increase on VQAv2. Notably, our around  1% gains over Wanda, the current state-of-the-art pruning method, is  promising advancements in this area **(Recognized by Reviewer of38, s93u, gYqu, and VeXf)**.
>
> **(2)** To ensure fair comparisons, we follow standard settings like 50% sparsity, where baseline results are strong, making additional gains challenging.
>
> **(3)** Our experiments in Table 8 at higher sparsity levels (65% ~ 80%) on LLaMA-1 show our method's ability to achieve substantial improvements.
>
> **(4)** Additional results in the rebuttal PDF **(See our general response)** further demonstrate that our method can achieve substantial gains, with improvements ranging from **1% ~ 7% at 60% sparsity, 2% ~ 7.6% at 70% sparsity, and 1.4% ~ 4.2% at 2:4 sparsity**.
>
> > **Q3: About performance in 2:4 sparsity setting.**
>
> **A3**: Following the suggestions, we evaluate our DSA under 2:4 sparsity on LLaVA (See Table below) and LLaMA-2  (See our general response). These consistent gains **(2.2% ~ 2.8% on LLaVA and 1.4% ~ 4.2% on LLaMA-2)**  across different model sizes and datasets suggest that our approach is more effective at maintaining performance under tighter sparsity constraints.
>
> | 7B LLaVA-1.5 | VQAv2 | SQA   | VQA   | 13B LLaVA-1.5 | VQAv2 | SQA   | VQA   |
> | ------------ | ----- | ----- | ----- | ------------- | ----- | ----- | ----- |
> | Dense        | 78.50 | 66.80 | 58.20 | Dense         | 80.00 | 74.94 | 61.30 |
> | Wanda (2:4)  | 68.92 | 55.06 | 45.42 | Wanda (2:4)   | 75.39 | 64.89 | 52.52 |
> | **Ours (2:4)**   | **71.18** | **57.44** | **48.25**| **Ours (2:4)**    | **76.75** | **67.13** | **54.05** |
>
> **Finally,** we hope our response could address the concerns, and we thank the reviewer again for the helpful comments. We are glad to discuss further comments and suggestions.

---

> ### Author Response · Authors · 2024-08-11
> **Look Forward to The Post-Rebuttal Feedback**
>
> Dear Reviewer 1f5c,
>
> We sincerely appreciate your thoughtful and constructive feedback. We have diligently addressed each of your concerns in our point-by-point rebuttal.  We hope our response has alleviated your concerns and illuminated the value of our work. If so, we would be immensely grateful if you could reconsider your recommendation. We assure you that all your insightful comments will be meticulously incorporated into the final manuscript. We have earnestly integrated feedback from all four reviewers and hope this is evident in your evaluation. Thank you for dedicating your valuable time to review our response.
>
> With deepest gratitude,
>
> Paper 66 Authors

---

> ### Author Response · Authors · 2024-08-12
> **Request to review the rebuttal [Author-Reviewer discussion phase ending soon]**
>
> Dear Reviewer 1f5c,
>
> We sincerely thank you for your valuable feedback, which has provided our paper with deeper insights. We promise to thoroughly reflect all your comments in the final manuscript. As we are towards the end of the author-reviewer discussion period, we request you to please go through our rebuttal, and we would be immensely grateful if you could reconsider your recommendation.
>
> Best regards,
>
> Paper 66 Authors

---

> > ### Author Response · Authors · 2024-08-13
> > **Request to review the rebuttal [only one day left]**
> >
> > Dear Reviewer 1f5c,
> >
> > Thank you for your time reviewing the paper. Your constructive feedback will help improve the quality of our paper.
> >
> > We have also addressed all your concerns in our rebuttal point-by-point. As we are towards the end of the author-reviewer discussion period, we request you to please go through our rebuttal, and we sincerely hope that reviewer will consider improving the recommendation.
> >
> > We thank you again for your time!
> >
> > Best regards,
> >
> > Paper 66 Authors

---

> > > ### Comment · Reviewer_1f5c · 2024-08-14
> > >
> > > Thanks to the authors for the detailed response.
> > >
> > > The new evaluation of structured pruning shows the effectiveness of the proposed method better. For the clarification part, I tried to spend more time better understanding the method design principle with the paper and response, but I still feel confused about the entire pipeline. While I acknowledge the contributions of this work, I am not a specialist in this domain and may miss some information here. After carefully considering your rebuttal and the feedback provided by other reviewers, I have decided to keep my initial rating.

---

### Author Rebuttal · Authors · 2024-08-06

# **General Response**


**Dear Reviewers, Area Chairs, Senior Area Chairs and Program Chairs,**

We sincerely thank all reviewers for their positive feedback and constructive comments. **In the initial review, 3 Positive ratings are given.** Reviewers positively acknowledge **the novelty of the idea, the methodology employed, the extensive experiments conducted, the superior performance, and the good presentation of the paper**. More encouragingly, **Reviewer of38, 1f5c, VeXf, and gYqu** think our **novel automated** **framework tackles an important problem in LLM efficiency** for the community.

**[Important problem]:**

- **Reviewer VeXf:** "tackles an important problem in LLM efficiency"

- **Reviewer of38:** "tackles an important problem"

**[Novelty]:**

- **Reviewer 1f5c**: "proposes a novel method"
- **Reviewer VeXf**: "novel automated framework"
- **Reviewer gYqu**: "framework for automatically discovering effective sparsity allocation functions"

**[Theoretical Soundness]:**
- **Reviewer s93u**: "technically sound and straightforward in principle"

**[Extensive experiments]:**

- **Reviewer of38**: "evaluated on a wide range of models"

- **Reviewer VeXf:** "Extensive experiments"

**[Superior performance]:**

- **Reviewer 1f5c:** "shows better performance than SOTA pruning methods"
- **Reviewer of38:** "demonstrated strong results compared to state-of-the-art methods"
- **Reviewer s93u:** "Empirical results demonstrate the strength of this approach"
- **Reviewer gYqu**: "demonstrates consistent performance improvements"
- **Reviewer VeXf:** "DSA outperforms existing methods"

**[Good presentation]:**

- **Reviewer 1f5c:** "insight of the paper is clear and easy to understand"
- **Reviewer s93u:** "This manuscript is a qualified paper"

In the past days, we carefully improved the experiments (using all computational resources we have), the clarifications, and the discussions of our work to address the concerns, the questions, and the requests of all four reviewers.

**In the attached rebuttal PDF, we provide detailed experiment results of higher sparsity ratios and 2:4  sparsity on LLaMA-2 and LLaMA-3 (Mean accuracies are summarized in Table below).** Our DSA method consistently boosts mean accuracies on seven zero-shot tasks across various LLaMA models and sparsity levels. The application of DSA leads to notable improvements in performance, **with improvements  ranging from 2.34% to 7.03% in Magnitude pruning, 1.95% to 7.68% in SparseGPT, and 1.92% to 3.30% in Wanda**, demonstrating the effectiveness of DSA in enhancing model accuracy in different scenarios and with various pruning methods.

*Table: Mean accuracies (%) of our DSA on 7 zero-shot task.*

| Method              | LLaMA-2-7B (60%)  | LLaMA-2-7B (70%)  | LLaMA-2-13B (60%) | LLaMA-2-13B (70%) | LLaMA-3-70B (60%) | LLaMA-3-70B (70%) | LLaMA-2-7B (2:4)  | LLaMA-2-7B (2:4)  |
| ------------------- | ----------------- | ----------------- | ----------------- | ----------------- | ----------------- | ----------------- | ----------------- | ----------------- |
| Magnitude           | 35.61             | 50.81             | 38.38             | 51.16             | 55.86             | 38.76             | 45.58             | 49.89             |
| **Magnitude (DSA)** | **37.95 (2.34↑)** | **57.84 (7.03↑)** | **46.06 (7.68↑)** | **54.28 (3.12↑)** | **60.24 (4.38↑)** | **42.98 (4.22↑)** | **49.78 (4.20↑)** | **53.38 (3.49↑)** |
| SparseGPT           | 43.61             | 60.68             | 48.76             | 70.14             | 65.03             | 43.22             | 50.94             | 54.86             |
| **SparseGPT (DSA)** | **45.56 (1.95↑)** | **61.31 (0.63↑)** | **50.04 (1.28↑)** | **72.12 (1.98↑)** | **67.34 (2.31↑)** | **45.73 (2.51↑)** | **52.66 (1.72↑)** | **56.35 (1.49↑)** |
| Wanda               | 36.08             | 60.90             | 41.46             | 72.00             | 40.51             | 40.44             | 48.75             | 55.03             |
| **Wanda (DSA)**     | **38.00 (1.92↑)** | **62.08 (1.18↑)** | **43.18 (1.72↑)** | **73.24 (1.24↑)** | **42.74 (2.23↑)** | **42.78 (2.34↑)** | **52.05 (3.30↑)** | **57.07 (2.04↑)** |

**Finally**, based on the constructive comments by all reviewers and our responses, **we will carefully revise the manuscript of our work**. We hope our detailed responses help address the concerns, the questions, and the requests of all  reviewers.

---

### Comment · Area_Chair_utcw · 2024-08-12
**Discussion period ending**

Dear Reviewers,

The author-reviewer discussion period will end within two days (Aug 13). Please respond to/acknowledge the author rebuttal and indicate whether it addressed your concerns. Thank you

Best,\
Area Chair

---

### Author Response · Authors · 2024-08-14
**Summary of Discussion Period and Thank all Reviewers**

**Dear Reviewers, Area Chairs, Senior Area Chairs, and Program Chairs,**

We are profoundly grateful for your exceptional dedication, insightful comments, and invaluable suggestions throughout this review process. Your thoughtful feedback has been instrumental in shaping and elevating our work.





We are deeply heartened to see that our responses have effectively addressed the major concerns raised by all reviewers. We are pleased to note that the all reviewers satisfied with the responses we provided after the discussion and **4 Accept ratings (2 Weak Accept & 2 Borderline accept) are given**. In particular, **Reviewer of38 and Reviewer gYqu, who are highly senior and professional scholars in our field (Confidence: 4), recognize our responses and have increased their final recommendations (2 Weak Accept ratings)**. In addition, we are grateful to Reviewers 1f5c and s93u for maintaining their positive stance on our work post-discussion.



While Reviewer VeXf has not responded to our additional rebuttal, we are confident that our comprehensive clarifications and comparison experiments have thoroughly addressed his concerns. We respectfully request that the Area Chair consider our extensive responses and the positive feedback from other reviewers when evaluating Reviewer VeXf's initial comments.



We want to express our deepest appreciation to all reviewers for their engagement during the discussion period. Your constructive feedback has been invaluable, and we are committed to incorporating your suggestions into our revised manuscript. Your insights will significantly enhance the quality and impact of our work.



We are truly thankful for the time and expertise you have invested in this process. Your contributions exemplify the highest standards of academic peer review and have pushed us to refine and strengthen our research.



With sincere gratitude and warmest regards,



Paper 66 Authors

---

### Decision · Program_Chairs · 2024-09-25

**Decision:**

Accept (poster)

**Comment:**

This paper receives ratings of 5,6,6,6,4. The majority of reviewers lean towards acceptance, and the reviewer who gave 4 seemed reasonably content with the rebuttal. The paper seeks to automate the process of assigning sparsity ratios for each layer in pruning LLMs, which is an important yet sometimes overlooked problem. The reviewers mentioned the method is novel, the experiments are extensive, and the results generally show improvement. Congratulations to authors for the paper acceptance.